# Structure of the Dicer-2–R2D2 heterodimer bound to a small RNA duplex

Sonomi Yamaguchi[1], Masahiro Naganuma[2,3], Tomohiro Nishizawa[4], Tsukasa Kusakizako[1], Yukihide Tomari[3,5 ✉], Hiroshi Nishimasu[1,6,7 ✉] & Osamu Nureki[1 ✉]

In flies, Argonaute2 (Ago2) and small interfering RNA (siRNA) form an RNA-induced silencing complex to repress viral transcripts[1]. The RNase III enzyme Dicer-2 associates with its partner protein R2D2 and cleaves long double-stranded RNAs to produce 21-nucleotide siRNA duplexes, which are then loaded into Ago2 in a defined orientation[2–5]. Here we report cryo-electron microscopy structures of the Dicer-2–R2D2 and Dicer-2–R2D2–siRNA complexes. R2D2 interacts with the helicase domain and the central linker of Dicer-2 to inhibit the promiscuous processing of microRNA precursors by Dicer-2. Notably, our structure represents the strand-selection state in the siRNA-loading process, and reveals that R2D2 asymmetrically recognizes the end of the siRNA duplex with the higher base-pairing stability, and the other end is exposed to the solvent and is accessible by Ago2. Our findings explain how R2D2 senses the thermodynamic asymmetry of the siRNA and facilitates the siRNA loading into Ago2 in a defined orientation, thereby determining which strand of the siRNA duplex is used by Ago2 as the guide strand for target silencing.

The specialized RNase III enzyme Dicer has a central role in the production of small RNAs, such as siRNAs and microRNAs[6,7] (miRNAs). *Drosophila* has two Dicer enzymes, Dicer-1 and Dicer-2, which associate with their double-stranded RNA (dsRNA)-binding proteins Loqs-PB and R2D2 and cleave short hairpin miRNA precursors (pre-miRNAs) and long dsRNA substrates to produce miRNA and siRNA duplexes, respectively[2,8]. Subsequently, miRNA duplexes are loaded into Ago1 and induce deadenylation, decay and/or translational repression of their endogenous mRNA targets[9,10]. By contrast, siRNA duplexes are loaded into Ago2 and facilitate nucleolytic cleavage of genetic invaders, such as viral transcripts and transposable elements[11–16].

The Dicer-2–R2D2 heterodimer has critical roles in both siRNA production and siRNA loading onto Ago2. Dicer-2–R2D2 processively cleaves long dsRNA substrates in an ATP-dependent manner to produce 21-nucleotide (nt) siRNA duplexes[4,5]. Subsequently, Dicer-2–R2D2 re-associates with an siRNA duplex, which is then loaded into Ago2 with the aid of the Hsc70/Hsp90 chaperone machinery[17–21]. R2D2 contributes to determining the specificities in the siRNA production and siRNA loading. Dicer-2 processes pre-miRNAs inaccurately in vitro, but R2D2 inhibits the promiscuous pre-miRNA processing by Dicer-2 (ref. [5]). Dicer-2–R2D2 efficiently binds highly paired siRNA duplexes, but not miRNA duplexes with central mismatches, thereby preventing the inappropriate loading of miRNA duplexes into Ago2 (ref. [22]). Notably, Dicer-2–R2D2 binds an siRNA duplex in a fixed orientation: the more thermodynamically stable 5′ end of the siRNA duplex is located near R2D2, whereas the other 5′ end with the weaker base-pairing stability is positioned near Dicer-2 (ref. [3]). In general, Ago2 uses the siRNA strand with the less thermodynamically stable 5′ end as the guide strand for target silencing[23,24], whereas the

other strand in the siRNA duplex is cleaved by Ago2 and discarded as the passenger strand[25]. Thus, the Dicer-2–R2D2 heterodimer senses the siRNA thermodynamic asymmetry and transfers the siRNA duplex into Ago2 in a defined orientation, thereby determining which strand of the siRNA duplex is used by Ago2 as the guide strand.

Previous structural studies of the Dicer enzymes from *Giardia intestinalis*, human (Dicer), *Drosophila* (Dicer-2) and *Arabidopsis* (DCL1 and DCL3) provided insights into their substrate recognition and cleavage mechanisms[26–31]. However, it remains unknown how Dicer-2–R2D2 selectively cleaves dsRNA substrates to produce siRNA duplexes, senses the siRNA thermodynamic asymmetry, and facilitates the loading of an siRNA duplex onto Ago2 in a fixed orientation. In this study, we solved the high-resolution cryo-electron microscopy (cryo-EM) structure of the Dicer-2–R2D2–siRNA complex, and provide mechanistic insights into dsRNA cleavage and siRNA loading by the Dicer-2–R2D2 heterodimer.

## Overall structure

An asymmetric *let-7*-derived siRNA can be loaded by Dicer-2–R2D2 into Ago2 in a fixed orientation in vitro[21]. To reconstitute the Dicer-2–R2D2–siRNA complex in the pre-loading state, we mixed the purified Dicer-2–R2D2 heterodimer with the *let-7* siRNA duplex, and then purified the ternary complex on a gel-filtration column. We determined the cryo-EM structures of the Dicer-2–R2D2 heterodimer and the Dicer-2–R2D2–siRNA complex at 3.3 Å resolution (Extended Data Fig. 1a–h and Supplementary Table 1).

The present structures illuminated the detailed architectures of the individual domains and interdomain linkers of Dicer-2, which were not

[1]Department of Biological Sciences, Graduate School of Science, The University of Tokyo, Tokyo, Japan. [2]RIKEN Center for Biosystems Dynamics Research, Yokohama, Japan. [3]Laboratory of RNA Function, Institute for Quantitative Biosciences, The University of Tokyo, Tokyo, Japan. [4]Molecular Cellular Biology Laboratory, Yokohama City University, Graduate School of Medical Science, Yokohama, Japan. [5]Department of Computational Biology and Medical Sciences, Graduate School of Frontier Sciences, The University of Tokyo, Tokyo, Japan. [6]Research Center for Advanced Science and Technology, Structural Biology Division, The University of Tokyo, Tokyo, Japan. [7]Department of Chemistry and Biotechnology, Graduate School of Engineering, The University of Tokyo, Tokyo, Japan. ✉e-mail: tomari@iqb.u-tokyo.ac.jp; nisimasu@g.ecc.u-tokyo.ac.jp; nureki@bs.s.u-tokyo.ac.jp

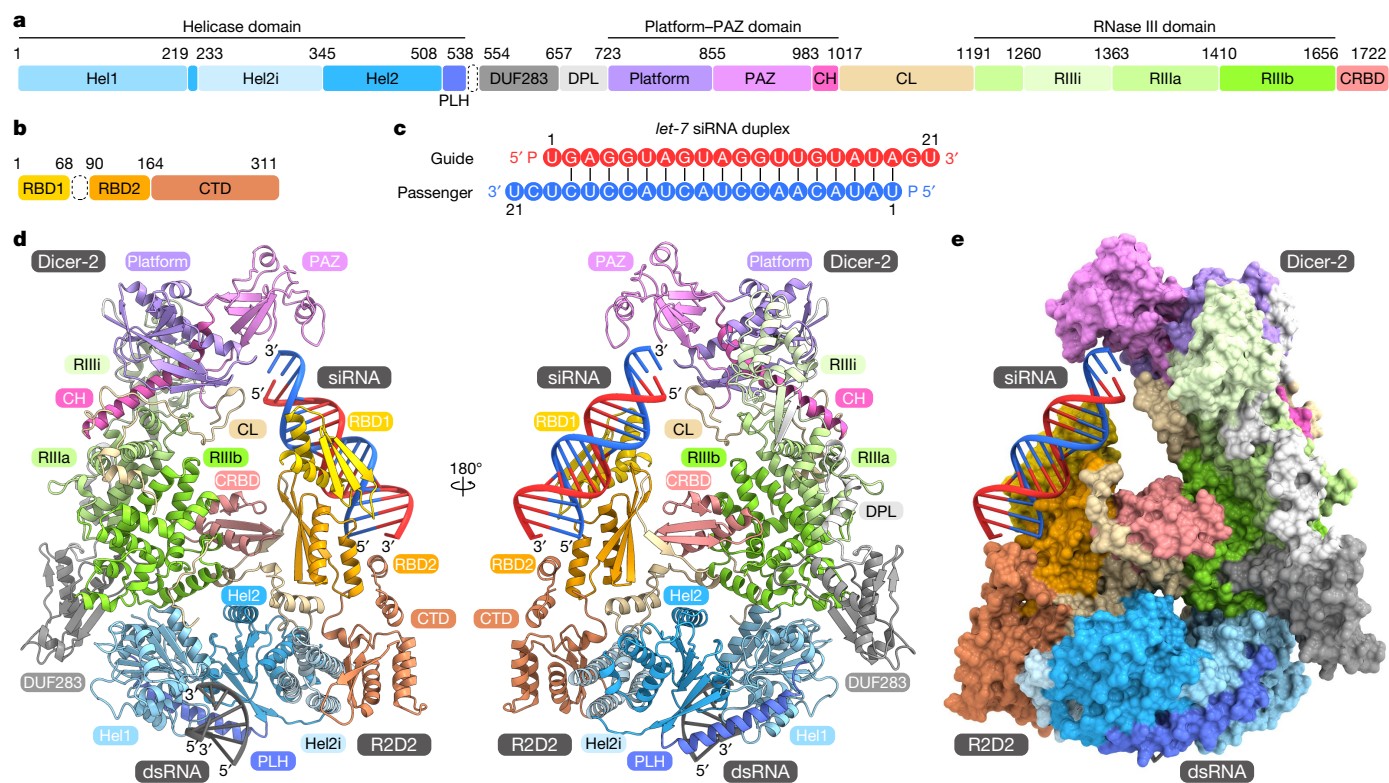

**Fig. 1 | Overall structure of the Dicer-2–R2D2–siRNA complex. a,b,** Domain organizations of Dicer-2 (**a**) and R2D2 (**b**). **c,** Nucleotide sequence of the *let-7* siRNA duplex. **d,e,** Overall structure of the Dicer-2–R2D2–siRNA complex. The Dicer-2–R2D2 heterodimer is shown as ribbon (**d**) and surface (**e**) models. PLH, pincer-like helix; DPL, DUF283–platform linker; CH, connector helix; CL, central linker.

visible in the previous cryo-EM reconstructions at approximately 7 Å resolution[28], and revealed that Dicer-2 comprises an amino-terminal helicase domain, a DUF283 domain, a platform–PAZ domain, two RNase III domains (RIIIa and RIIIb) and a carboxy-terminal dsRNA-binding domain (CRBD) (Fig. 1a–e and Supplementary Video 1). The helicase domain consists of the Hel1, Hel2 and Hel2i domains, and a pincer-like helix. The platform and PAZ domains are linked by a connector helix. The RIIIa and RIIIb domains form an intramolecular dimer to create the central RNase III active site. The RIIIa domain interacts with the connector helix and the DUF283–platform linker (Extended Data Fig. 2a), whereas the RIIIb domain interacts with the Hel1, DUF283 and CRBD domains (Extended Data Fig. 2b). Our high-resolution structures further revealed the presence of an α-helical domain inserted within the RIIIa domain (referred to as RIIIi) and a prominent linker region between the PAZ and RIIIa domains (referred to as the central linker) (Fig. 1a,d). The RIIIi domain interacts with the DUF283–platform linker and the platform domain (Extended Data Fig. 2c). Notably, the central linker is mostly ordered and extensively interacts with the eight domains (Hel1, Hel2i, Hel2, platform, PAZ, RIIIa, RIIIb and CRBD) of Dicer-2 and R2D2 (Fig. 1d and Extended Data Fig. 2d–f). The central linker regions are highly conserved among the Dicer-2 orthologues, but not the miRNA-producing Dicers (human Dicer and *Drosophila* Dicer-1) (Extended Data Fig. 3a,b). R2D2 comprises two dsRNA-binding domains (RBD1 and RBD2) and a carboxy-terminal domain (CTD) (Fig. 1b,d). The three domains adopt dsRNA-binding domain folds with an αβββα topology.

The Dicer-2–R2D2–siRNA structure revealed two RNA duplex molecules: one bound to the helicase domain of Dicer-2 and the other bound to R2D2 (Fig. 1c–e, Extended Data Fig. 4a). The Dicer-2-bound RNA duplex was not well resolved in the density map (Extended Data Fig. 4b), suggesting that it does not bind stably to the helicase domain of Dicer-2. Since the density was ambiguous but fitted to the unstable end

relative to the stable end of the RNA duplex, we modelled the nucleotides at the unstable end (nucleotides g1–5 and p15–21 in Fig. 1c) into the density. Nonetheless, the guide and passenger strands cannot be functionally defined at the dicing step[17], so we do not discriminate between the two strands hereafter. By contrast, the R2D2-bound RNA duplex (except for nucleotides g21 and p21) was well resolved in the density map (Extended Data Fig. 4c), enabling us to unambiguously model the guide and passenger strands. These observations indicate that the RNA molecules bound to Dicer-2 and R2D2 represent a dsRNA substrate at the initial recognition state in the dicing process and an siRNA product at the strand-selection state in the loading process, respectively. Thus, we refer to the RNA molecules bound to Dicer-2 and R2D2 as dsRNA and siRNA, respectively.

## Structural changes upon siRNA binding

A structural comparison of Dicer-2–R2D2 and Dicer-2–R2D2–siRNA revealed that, although their overall structures are similar, the central linker becomes ordered and interacts with Dicer-2 CRBD and R2D2 RBD2 upon siRNA binding (Extended Data Fig. 5a–c). R2D2 RBD1 in the Dicer-2–R2D2 structure was not resolved in the density map (Extended Data Figs. 1g and 5a), suggesting that RBD1 is highly mobile in the siRNA-unbound state. By contrast, RBD1 becomes ordered and interacts with RBD2 in the Dicer-2–R2D2–siRNA structure (Extended Data Figs. 1h and 5b–d), indicating a structural change in R2D2 upon siRNA binding.

## Interaction between Dicer-2 and R2D2

The Dicer-2 Hel2i domain interacts with the R2D2 CTD domain through hydrophobic and electrostatic interactions (Fig. 2a), consistent with

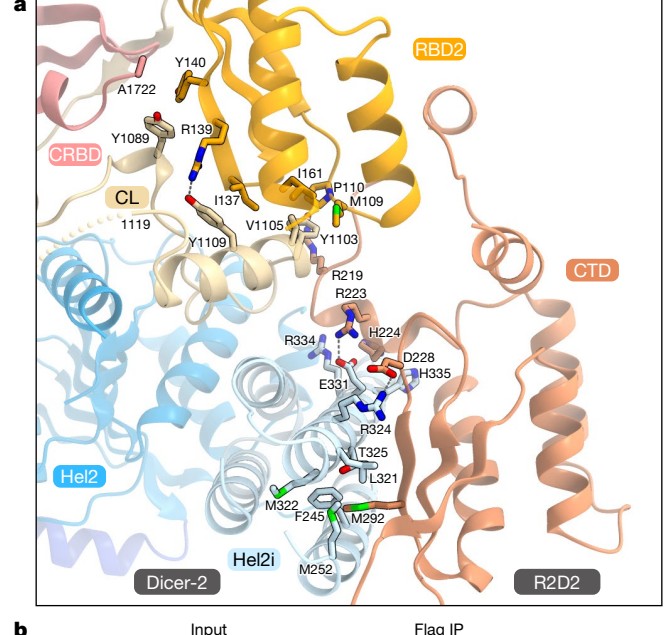

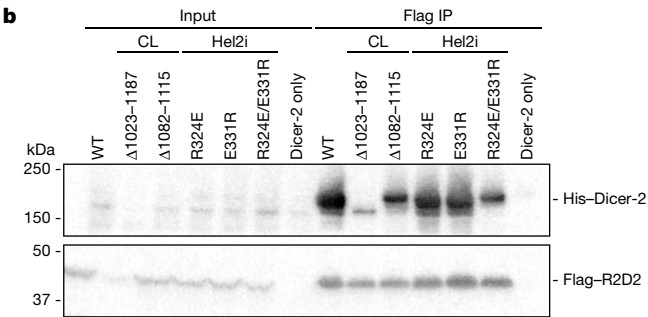

**Fig. 2 | Interaction between Dicer-2 and R2D2. a**, The interface between Dicer-2 and R2D2. Hydrogen bonds and electrostatic interactions are indicated by grey dashed lines. **b**, Pull-down experiments. His-tagged Dicer-2 (wild type or CL or Hel2i mutants) and Flag-tagged wild-type R2D2 were co-expressed in Sf9 cells and purified using anti-Flag beads (*n* = 3 independent experiments). The cell lysates and the bound proteins were analysed by western blotting using anti-Flag and anti-Dicer-2 antibodies. Δ1023–1187, the P1022-GGGS-P1188 mutant; Δ1082–1115, the E1081-GGGS-G1116 mutant.

previous studies indicating that Dicer-2 binds R2D2 via its helicase domain[32,33]. In particular, R324 and E331 of Dicer-2 form salt bridges with D288 and R223 of R2D2, respectively (Fig. 2a). Indeed, the R324E/E331R mutations substantially reduced the interaction between Dicer-2 and R2D2 (Fig. 2b). Furthermore, the central linker hydrophobically interacts with Dicer-2 Hel2/Hel2i/CRBD and R2D2 RBD2/CTD. Specifically, Y1089, Y1109, V1105 and Y1103 in the central loop interact with the RBD2 domain of R2D2 (Fig. 2a). The Dicer-2 mutant with a truncated central linker (residues 1082–1115) bound to R2D2 less efficiently (Fig. 2b), confirming the involvement of the central linker in R2D2 binding. The Dicer-2 mutant lacking the central linker (residues 1023–1187) was not expressed in insect cells as a soluble protein (Fig. 2b), suggesting that the central linker is essential for the structural integrity of the Dicer-2 protein.

## dsRNA recognition by Dicer-2

The helicase domain of Dicer-2 adopts a C-shaped structure similar to that of RIG-I[34], and contains a canonical ATP-binding site formed by P29–T35 (motif I), D139–C141 (motif II), and R494–R496 (motif VI) (Extended Data Fig. 6a–e), consistent with the ATPase activity of Dicer-2 (refs. [4,5]). F225 forms a hydrophobic core in the Hel2 domain (Extended Data Fig. 6f), explaining why the F225G mutation abolished Dicer-2-mediated siRNA production[28]. RIG-I contains a carboxy-terminal regulatory domain, which is connected to the Hel2 domain with the V-shaped pincer helix (Extended Data Fig. 6c). By contrast, the helicase domain of Dicer-2 lacks the carboxy-terminal regulatory domain and has a

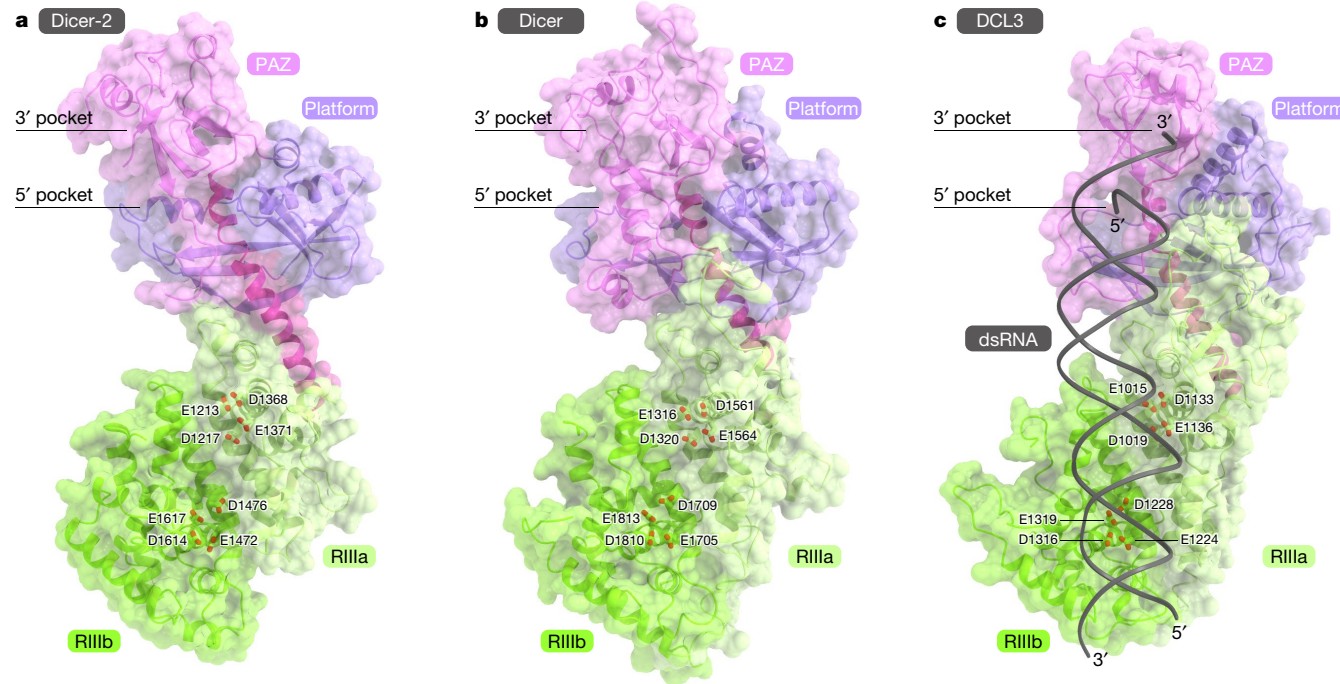

**Fig. 3 | Active site of Dicer-2. a–c**, Platform–PAZ and RNase III domains of Dicer-2 (**a**), human Dicer (PDB: 5ZAL) (**b**) and plant DCL3 (PDB: 7VG3) (**c**).

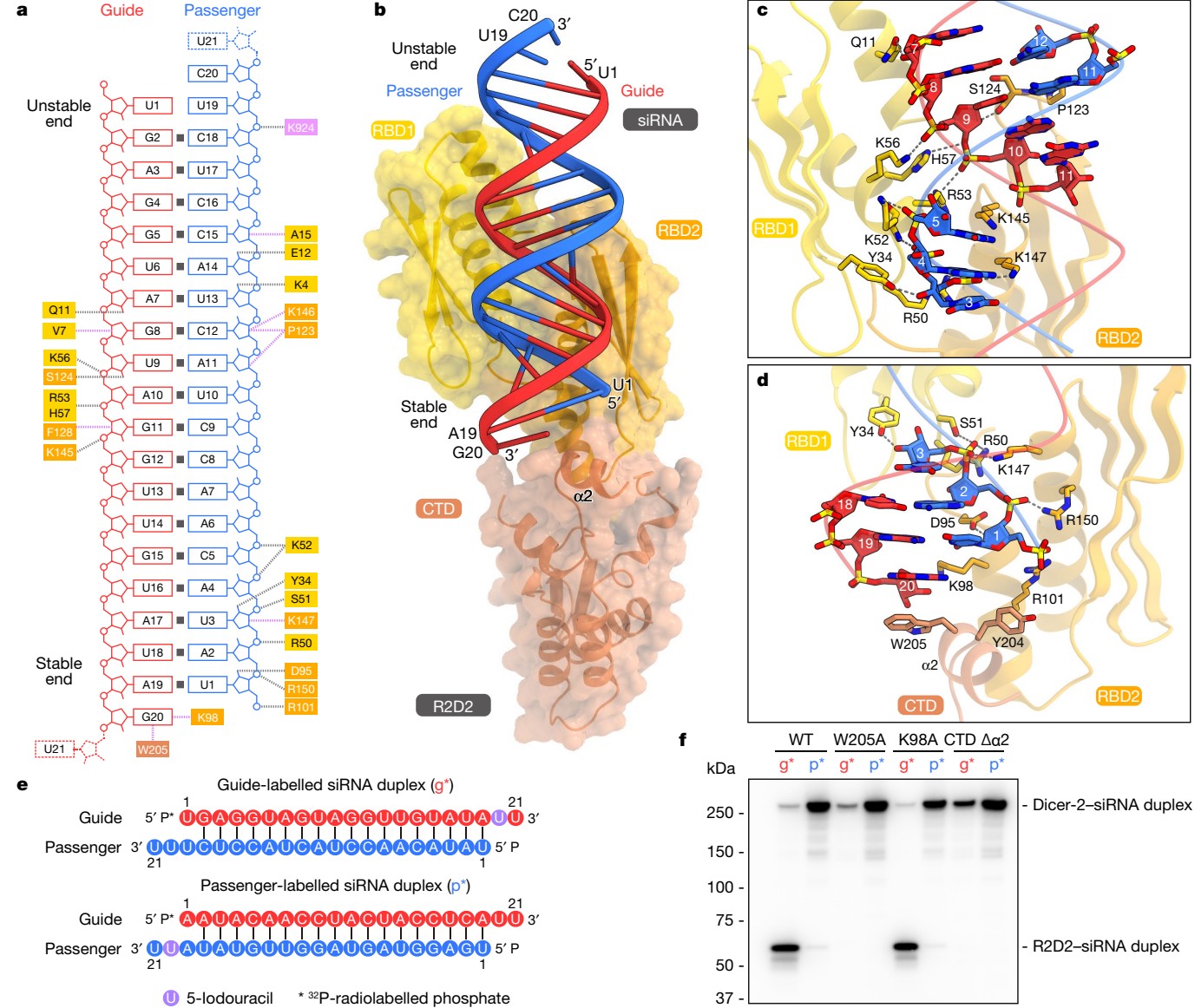

**Fig. 4 | siRNA recognition by R2D2. a**, Schematic of siRNA recognition by R2D2. The disordered nucleotides are depicted as dashed lines. **b**, Structure of R2D2 bound to siRNA. **c,d**, Recognition of the central region (**c**) and the stable end (**d**) of siRNA by R2D2. **e**, Nucleotide sequences of siRNA duplexes used in the photocrosslinking experiments in **f**. **f**, Photocrosslinking experiments.

Dicer-2 and R2D2 (WT or mutants) were incubated with 5′-radiolabelled siRNA (g* or p*) bearing 5-iodouracil at position 20. The reaction mixture was analysed by SDS–PAGE, and crosslinked proteins were detected using phosphorimaging. CTD Δα2, the R2D2 mutant in which F201 and H215 are connected by a GGGS linker. *n* = 3 independent experiments.

shorter pincer-like helix (Extended Data Fig. 6b). In the Dicer-2–R2D2–siRNA structure, dsRNA is recognized by V67, G90, H147 and K177 in the Hel1 domain of Dicer-2 in a sequence-independent manner (Extended Data Fig. 6g), similar to the blunt-end dsRNA in the Dicer-2–dsRNA structure[28] (Extended Data Fig. 6h). These structural observations are consistent with previous studies indicating that the Dicer-2 helicase domain initially recognizes both long dsRNA substrates with a blunt end and a 2-nt 3′-overhanging end in the dicing process[28,35,36].

## dsRNA cleavage by Dicer-2

The Dicer enzymes recognize the 5′-monophosphate and 3′-overhang of dsRNA substrates, using a basic pocket in the platform domain (5′-pocket) and a hydrophobic pocket in the PAZ domain (3′-pocket), respectively[27,30,31]. The present structure revealed that Dicer-2 has both 5′- and 3′-pockets similar to those of human Dicer and *Arabidopsis* DCL3 (Fig. 3a–c and Extended Data Fig. 7a–f). A previous mutational analysis

indicated that H743 and R943 in the 5′-pocket are involved in siRNA production[37].

The RNase III domain of Dicer-2 is structurally similar to those of human Dicer[29] and *Arabidopsis* DCL3[31], and contains the active sites formed by conserved acidic residues (E1213, D1217, D1368 and E1371 in RIIIa and E1472, D1476, D1614 and E1617 in RIIIb) (Fig. 3a–c). A structural comparison of Dicer-2–R2D2 with DCL3–dsRNA suggested that Dicer-2 recognizes the 5′-monophosphate of dsRNA substrates in the 5′-pocket and cleaves the dsRNAs 21 nt away from the 5′ end in the RIIIb active site (Extended Data Fig. 8a,b), consistent with a previous proposal[37]. The modelled dsRNA sterically clashes with the helicase and CRBD domains of Dicer-2 (Extended Data Fig. 8b), suggesting that these domains undergo structural rearrangements upon the binding of dsRNA substrates. Supporting this notion, in the Dicer-2 structure predicted by AlphaFold2 (ref. [38]), the helicase domain is arranged similarly to that in the DCL1–dsRNA structure[30] and interacts with the DUF283 domain (Extended Data Fig. 8c,d). Furthermore, a comparison

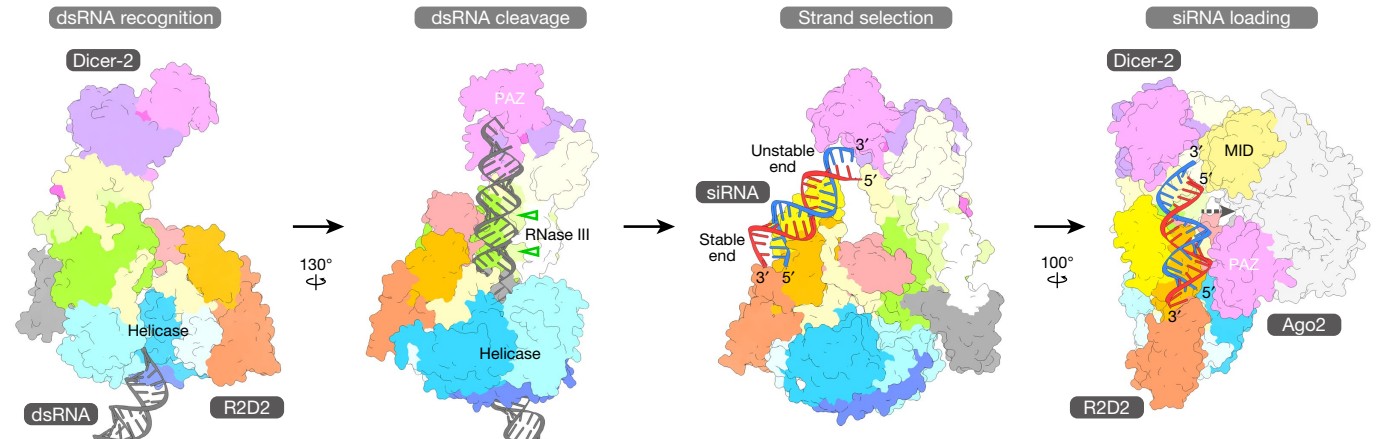

**Fig. 5 | Action mechanism of the Dicer-2–R2D2 heterodimer.** The structural model of the RISC-loading complex was built by manually docking the Ago2 structure (PDB: 6MFR) onto the Dicer-2–R2D2–siRNA complex. The Hsc70/Hsp90 chaperone machinery has been omitted for clarity.

of Dicer-2–R2D2 with human Dicer–TRBP–pre-miRNA suggested that R2D2 could sterically clash with a modelled pre-miRNA substrate (Extended Data Fig. 9a–c), explaining why R2D2 inhibits promiscuous pre-miRNA processing by Dicer-2 (ref. [5]).

## siRNA strand selection by R2D2

The present structure revealed that R2D2 fixes the siRNA duplex in a defined orientation (Fig. 4a,b and Extended Data Fig. 10a). The central region (nucleotides g5–15 and p5–15) and the stable end (nucleotides g16–20 and p1–4) of siRNA are recognized by R2D2. By contrast, the unstable end (nucleotides g1–4 and p16–20, containing a U–U pair) is not recognized by either the 3′- or 5′-pocket of Dicer-2, and instead is exposed to the solvent. The central region of siRNA is extensively recognized by RBD1 (Q11, R50, K52, R53, K56 and H57) and RBD2 (S124, P123, K145 and K147) of R2D2 through sugar–phosphate backbone interactions (Fig. 4c), consistent with a previous study showing that Dicer-2–R2D2 preferentially binds an siRNA duplex without central mismatches[22]. Notably, the 1-nt 3′-overhang at the siRNA stable end is anchored by the RBD2 and CTD of R2D2 (Fig. 4d and Extended Data Fig. 10b). The 5′-phosphate group of the nucleotide p1 interacts with R101 and R150 of R2D2, consistent with a previous study showing that Dicer-2–R2D2 preferentially binds an siRNA duplex with a 5′-phosphate[3]. The ribose and nucleobase moieties of the nucleotide g20 stack with K98 and W205 of R2D2, respectively (Fig. 4d). Y204 in the second α-helix (α2) in CTD stacks with R101 in RBD2, stabilizing the RBD2–CTD interface.

To validate our structural findings, we performed photocrosslinking assays, using siRNA duplexes containing 5-iodouracil at position 20 of the guide strand (g*) or the passenger strand (p*) (Fig. 4e). The guide and passenger strands were crosslinked to R2D2 and Dicer-2, respectively (Fig. 4f), as in a previous study[3], consistent with our structural finding that the stable and unstable ends of the bound siRNA duplex are located in the vicinities of R2D2 (CTD) and Dicer-2 (PAZ), respectively. Whereas the K98A mutation did not affect the crosslinking with the siRNA duplexes, the W205A mutation and the CTD α2 deletion abolished the crosslinking of g* to R2D2, but not that of p* to Dicer-2 (Fig. 4f), confirming that the siRNA stable end is located near W205. Notably, the CTD α2 deletion increased the crosslinking of g* to Dicer-2 (Fig. 4f). These results indicated that more siRNA duplexes bind to the CTD α2 deletion mutant in the opposite orientation, thereby highlighting the contribution of the CTD α2 to the asymmetrical siRNA binding. An siRNA duplex with a 1-nt 3′-overhang was similarly crosslinked to R2D2 (Extended Data Fig. 10c), consistent with the observation that U21 of the siRNA is disordered and not recognized by R2D2 in the present

structure (Extended Data Fig. 10b). A blunt-end siRNA duplex was also crosslinked to R2D2 (Extended Data Fig. 10c). Consistently, the terminal base pair of a modelled blunt-end siRNA stacks with W205 and K208 of R2D2 (Extended Data Fig. 10d). These results indicated that R2D2 recognizes the double-helical conformation, rather than the 3′-overhang structure, of an siRNA duplex in the strand-selection process.

Together, our structural and functional data revealed that R2D2 prefers to bind the double-helical conformation at the end of an siRNA duplex in a sequence-independent manner. Consequently, the more thermodynamically stable end (with greater double-helical character) of the siRNA duplex is preferentially anchored by R2D2 in equilibrium, leading to the asymmetric recognition of the siRNA duplex by the Dicer-2–R2D2 heterodimer.

## Discussion

We determined the high-resolution structure of the Dicer-2–R2D2 heterodimer bound to two RNA duplexes, which represent a dsRNA substrate in the pre-dicing, initial recognition state and an siRNA product in the pre-loading, strand-selection state. The structure provided mechanistic insights into dsRNA substrate recognition and siRNA thermodynamic asymmetry sensing by Dicer-2–R2D2. On the basis of the present structure, along with previous functional data, we propose a model of siRNA production and siRNA loading by the Dicer-2–R2D2 heterodimer (Fig. 5 and Supplementary Video 2). The helicase domain of Dicer-2 recognizes a long dsRNA substrate and then undergoes a conformational change. The dsRNA substrate passes through the helicase domain, and the 5′ end of the dsRNA is anchored by the 5′-pocket in the platform–PAZ domain. The dsRNA substrate is cleaved in the RNase III active site, yielding 21-nt siRNA duplexes. The produced siRNA duplex is released from the active site, and then recaptured by R2D2. While the thermodynamically stable end of the siRNA duplex is recognized by R2D2, the 5′-phosphate of the siRNA guide strand is exposed to the solvent. Notably, Ago2 uses the MID domain to recognize the 5′-phosphate of the siRNA guide strand[39,40], and the Hsc70–Hsp90 chaperone machinery facilitates the docking of Ago2 on the Dicer-2–R2D2–siRNA complex in a manner dependent on the recognition of the 5′-phosphate of the siRNA guide strand[21]. These observations suggest that Ago2 adopts an open conformation by the action of the Hsc70–Hsp90 chaperone machinery, and captures the 5′-phosphate of the guide strand in the siRNA duplex bound to Dicer-2–R2D2. In this way, the Dicer-2–R2D2 heterodimer senses the siRNA thermodynamic asymmetry and facilitates siRNA loading into Ago2 in a fixed orientation, thereby determining which strand of the siRNA duplex is used by Ago2 as the guide strand for target silencing. Future research should focus on the structural

elucidation of the Dicer-2–R2D2–siRNA–Ago2 quaternary complex for a complete understanding of the RNA-induced silencing complex (RISC) assembly mechanism.

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

# Methods

## Expression and purification of the Dicer-2–R2D2 heterodimer

Dicer-2 and R2D2 were co-expressed in Sf9 insect cells using the Bac-to-Bac system (Invitrogen). The gene encoding Dicer-2 (residues 1–1722) was cloned into a modified pFastBac vector (Invitrogen), in which the N-terminal 6×His tag was replaced with an 8×His–GFP tag. Dicer-2 was also cloned into the modified pFastBac vector, in which the N-terminal 6×His tag and the following TEV protease cleavage site were replaced with an 8×His tag, to improve the yield of the purified protein. The gene encoding R2D2 (residues 1–311) was cloned into a modified pFastBac vector, in which the N-terminal 6×His tag was replaced with a 3×Flag tag. The sequences of the DNA oligonucleotides used for the vector construction are listed in Supplementary Table 2.

For the preparation of the Dicer-2–R2D2 heterodimer, Sf9 cells at a cell density of $3.0 \times 10^6$ cells per ml in Sf900 II medium (Thermo Fisher Scientific) were co-infected with the baculoviruses expressing 8×His–GFP-tagged Dicer-2 and 3×Flag-tagged R2D2 and incubated for 72 h at 27 °C. The cells were then collected and disrupted in lysis buffer (30 mM Hepes-KOH, pH 7.4, 100 mM potassium acetate, 2 mM magnesium acetate, 0.5% NP-40, and 5% glycerol). The lysate was centrifuged at 40,000$g$ for 30 min, and the supernatant was incubated with anti-Flag M2 affinity resin (Sigma-Aldrich) for 1 h. The resin was washed with wash buffer (30 mM Hepes-KOH, pH 7.4, 800 mM NaCl, 2 mM magnesium acetate, 1% Triton-X 100, and 5% glycerol), and the protein was then eluted with elution buffer (30 mM Hepes-KOH, pH 7.4, 300 mM NaCl, 2 mM MgCl$_2$, 0.1 mg ml$^{-1}$ 3×Flag peptide, 20 mM imidazole, and 5% glycerol). The eluted protein was incubated with Ni-NTA Agarose resin (Qiagen), and the protein was eluted with elution buffer (30 mM Hepes-KOH, pH 7.4, 300 mM NaCl, 2 mM MgCl$_2$, 300 mM imidazole and 5% glycerol). The eluted protein was treated with TEV protease and dialysed against dialysis buffer (30 mM Hepes-KOH, pH 7.4, 300 mM NaCl, 2 mM MgCl$_2$, 40 mM imidazole and 5% glycerol). The protein sample was passed through a Ni-NTA Agarose column, to remove the 8×His–GFP and TEV protease. The Dicer-2–R2D2 protein was further purified on a Superdex 200 10/300 Increase column (GE Healthcare), equilibrated in 30 mM Hepes-KOH, pH 7.4, 100 mM KCl, 2 mM MgCl$_2$, 1 mM DTT and 0.02% glycerol.

For the preparation of the Dicer-2–R2D2–siRNA complex, Sf9 cells were co-infected with the baculoviruses expressing 8×His-tagged Dicer-2 and 3×Flag-tagged R2D2, and the Dicer-2–R2D2 protein was purified using anti-Flag M2 affinity resin and Ni-NTA Agarose, as described above. The Dicer-2–R2D2 protein was further purified by chromatography on a HiLoad Superdex 200 16/600 column (GE Healthcare), equilibrated with 30 mM Hepes-KOH, pH 7.4, 100 mM potassium acetate, 2 mM magnesium acetate, 1 mM DTT, and 0.02% glycerol. The purified Dicer-2–R2D2 and the *let-7* siRNA duplex (Ajinomoto Bio-Pharma) were mixed at a 1:2 ratio at room temperature, and the Dicer-2–R2D2–siRNA complex was purified by chromatography on a Superdex 200 10/300 Increase column, equilibrated in 30 mM Hepes-KOH, pH 7.4, 100 mM potassium acetate, 2 mM magnesium acetate, 1 mM DTT and 0.02% glycerol.

## Cryo-EM sample preparation

The Dicer-2–R2D2 complex was concentrated to $A_{280} = 0.6$, using a Vivaspin centrifugal filter device (100 kDa MW cut-off, Sartorius). The sample (3 µl) was applied to a freshly glow-discharged Cu 300 mesh R1/1 grid (Quantifoil), in a Vitrobot Mark IV (FEI) at 4 °C, with a waiting time of 30 s and a blotting time of 4 s under 100% humidity conditions. The Dicer-2–R2D2–siRNA complex was concentrated to $A_{280} = 0.9$, using the Vivaspin centrifugal filter device. The sample (3 µl) was applied to a freshly glow-discharged Au 300 mesh R1/1 grid (Quantifoil), in a Vitrobot Mark IV at 4 °C, with a waiting time of 30 s and a blotting time of 4 s under 100% humidity conditions. The grids were plunge-frozen in liquid ethane cooled at liquid nitrogen temperature.

## Cryo-EM data collection and processing

Cryo-EM data were collected using a Titan Krios G3i microscope (Thermo Fisher Scientific), running at 300 kV and equipped with a Gatan Quantum-LS Energy Filter (GIF) and a Gatan K3 Summit direct electron detector in the electron counting mode.

Micrographs for Dicer-2–R2D2 were recorded at a nominal magnification of ×105,000, corresponding to a calibrated pixel size of 0.83 Å at the electron exposure of 15.8 e$^-$ per pixel per s for 2.30 s, resulting in an accumulated exposure of 53 e$^-$ Å$^{-2}$. The data were automatically collected by the image shift method using the SerialEM software[41], with a defocus range of −1.6 to −0.8 µm, and 2,745 movies were obtained and processed using RELION-3.1. From the 2,745 motion-corrected and dose-weighted micrographs, 1,688,210 particles were initially picked, and extracted at a pixel size of 3.66 Å. These particles were subjected to several rounds of 2D and 3D classifications. The selected 324,630 particles were re-extracted at a pixel size of 1.25 Å, and then subjected to 3D refinement, per-particle defocus refinement, beam-tilt refinement, Bayesian polishing[42] and 3D classification with the mask focusing on Dicer-2 CRBD and R2D2. The selected 144,979 particles were subjected to 3D refinement, and subsequent postprocessing of the map improved its global resolution to 3.3 Å, according to the Fourier shell correlation (FSC) = 0.143 criterion[43]. The local resolution was estimated by RELION-3.1.

Micrographs for Dicer-2–R2D2–siRNA were recorded at a nominal magnification of ×105,000, corresponding to a calibrated pixel size of 0.83 Å at the electron exposure of 15 e$^-$ per pixel per s for 2.30 s, resulting in an accumulated exposure of 48 e$^-$ Å$^{-2}$. The data were automatically collected by the image shift method using the SerialEM software, with a defocus range of −1.6 to −0.8 µm. In total, 3,663 movies were obtained, and the beam-induced motion correction, dose-weighting and CTF estimation were conducted similarly to those for Dicer-2–R2D2. From the 3,663 motion-corrected and dose-weighted micrographs, 2,181,396 particles were initially picked, and extracted at a pixel size of 4.15 Å. These particles were subjected to several rounds of 2D and 3D classifications. The selected 179,826 particles were then re-extracted at a pixel size of 0.99 Å, and subjected to 3D refinement, per-particle defocus refinement, beam-tilt refinement and Bayesian polishing. The particles were again subjected to 3D refinement, and subsequent postprocessing of the map improved its global resolution to 3.3 Å, according to the FSC = 0.143 criterion.

## Model building and validation

The initial model of Dicer-2–R2D2 was built using Buccaneer[44], and the model was then manually built using COOT[45]. The model of the Dicer-2–R2D2–siRNA complex was built based on the Dicer-2–R2D2 model. The density maps were improved with the DeepEMhancer program[46]. The models were refined using Servalcat Refmac5 (ref. [47]), with external restraints prepared by ProSMART[48] and LIBG[49]. The structures were validated using MolProbity[50] from the PHENIX package. In the Dicer-2–R2D2 complex, residues 1–7, 90–94, 254–272, 346–353, 426–435, 539–553, 693–700, 955–968, 1052–1063, 1120–1129, 1146–1169, 1414–1421, 1564–1605, 1674–168, and 1721–1722 of Dicer-2, residues 1–2, 70–89, 164–187, 212–214 and 310 of R2D2 are not included in the final model, since these regions are not well resolved in the density map. In the Dicer-2–R2D2–siRNA complex, residues 1–7, 255–271, 345–351, 427–435, 539–553, 606–610, 693–700, 837–840, 955–968, 1042–1082, 1120–1129, 1146–1170, 1414–1421, 1564–1605 and 1673–1681 of Dicer-2, residues 1–94, 164–187 and 211–215 of R2D2, nucleotides g6–g21 and p1–p14 of the dsRNA, and nucleotides g21 and p21 of the siRNA are not included in the final model. The curves representing model versus full, half1 and half2 were calculated using Servalcat[47]. In brief, the final models were 'shaken' by introducing random shifts to the atomic coordinates with a root mean squared deviation of 0.3 Å, and were refined against the first half map. The statistics of the 3D reconstruction and model refinement

are summarized in Supplementary Table 1. The cryo-EM density maps were calculated with UCSF ChimeraX[51], and the molecular graphics were prepared with CueMol (http://www.cuemol.org).

## Pull-down experiments

The Dicer-2–R2D2 heterodimer (wild-type or mutant 8×His–Dicer-2 and 3×Flag–R2D2) was co-expressed in Sf9 cells (ATCC, catalogue no. CRL-1711), using the Bac-to-Bac system. The cells were collected by centrifugation, and then solubilized in lysis buffer (30 mM Hepes-KOH, pH 7.4, 100 mM potassium acetate, 2 mM magnesium acetate, 0.5% NP-40, and 5% glycerol). The lysate was centrifuged at 10,000$g$ for 15 min, and then half of the supernatant was incubated with anti-Flag M2 affinity magnetic beads for 1 h. The beads were washed with wash buffer (30 mM Hepes-KOH, pH 7.4, 800 mM NaCl, 2 mM magnesium acetate, 1% Triton-X 100, and 5% glycerol), and then with lysis buffer without NP-40. The beads were treated with SDS–PAGE sample buffer. The lysate and the bound proteins were analysed by 5–20% SDS–PAGE. Western blotting was performed at 20 V (constant voltage) for 30 min, using an Immobilon-P Transfer Membrane (Merck), an EzFastBlot HMW (ATTO), and a Trans-Blot Turbo Transfer System (Bio-Rad). After the transfer, the membranes were incubated on a tilting shaker with blocking buffer (TBST containing 1% skim milk) for 10 min, primary antibodies: anti-Flag antibody (1:2,000) and anti-Dicer-2 antibody (1:1,000) for 60 min, and horseradish peroxidase-conjugated secondary antibodies: goat anti-rabbit IgG (H+L) antibody (1:4,000) and goat anti-mouse IgG (H+L) antibody (1:4,000) for 30 min. The chemiluminescence was then detected using Luminata Forte Western HRP substrate (Merck) and an Amersham Imager 600 (GE Healthcare). R2D2 and Dicer-2 were detected with an anti-Flag-antibody (Sigma) and an anti-Dicer-2 antibody (Abcam), respectively.

## Photocrosslinking experiments

For photocrosslinking experiments, siRNA-1 and siRNA-4 were 5′-radiolabelled using T4 polynucleotide kinase, and then annealed with a 1.5-fold excess of siRNA-2 or siRNA-3 and siRNA-2 (1-nt overhang) or siRNA-5 (0-nt overhang), respectively (Supplementary Table 2). The 5-iodouracil-containing siRNA duplex (20 nM) and Dicer-2–R2D2 (40 nM) were incubated for 30 min. The samples (7 μl per well) were transferred to a Terasaki plate (Greiner BIO-ONE), and then exposed to > 300 nm UV radiation for 60 s, using a UV crosslinker (SP-11 Spot Cure, Ushio) with a uniform radiation lens (USHIO) and a long-path filter (300 nm, ASAHI SPECTRA), at 15 cm from the light. Aliquots of the reaction mixture were transferred into a new tube and mixed with SDS–PAGE sample buffer. The samples were analysed by 5–20% SDS–PAGE, and crosslinked proteins were detected by phosphorimaging.

## Reporting summary

Further information on research design is available in the Nature Research Reporting Summary linked to this paper.

## Data availability

The structural models and density maps have been deposited in the Protein Data Bank under the accession codes 7V6B (Dicer-2–R2D2) and 7V6C (Dicer-2–R2D2–siRNA). The raw images have been deposited in the Electron Microscopy Public Image Archive under the accession codes EMD-31741 (Dicer-2–R2D2) and EMD-31742 (Dicer-2–R2D2–siRNA).

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

**Acknowledgements** We thank W. Shihoya and K. Oogomori for assistance with the expression and purification of Dicer-2–R2D2; the staff scientists at The University of Tokyo's cryo-EM facility, especially H. Yanagisawa, A. Tsutsumi, M. Kikkawa and R. Danev, for help with cryo-EM data collection; K. Yamashita and A. Tomita for assistance with the structural determination; and H. Iwakawa and Y. Sakurai for technical assistance. This work was supported by Cabinet Office, Government of Japan, Public/Private R&D Investment Strategic Expansion Program (PRISM) Grant Number JPJ008000 (O.N.), CREST Grant Number JPMJCR20E2 (O.N.), JSPS KAKENHI Grant Numbers 21J11067 (S.Y.), 18H05271 (Y.T.) and 18H02384 (H.N.), and the ANRI Fellowship (S.Y.).

**Author contributions** S.Y. performed sample preparation, structural determination and biochemical analyses with assistance from M.N.; T.N. and T.K. performed cryo-EM data collection; H.N. performed structural analyses; S.Y. and H.N. wrote the manuscript with assistance from M.N., Y.T. and O.N.; Y.T., H.N. and O.N. supervised all of the research.

**Competing interests** The authors declare no competing interests.

**Additional information**
**Correspondence and requests for materials** should be addressed to Yukihide Tomari, Hiroshi Nishimasu or Osamu Nureki.

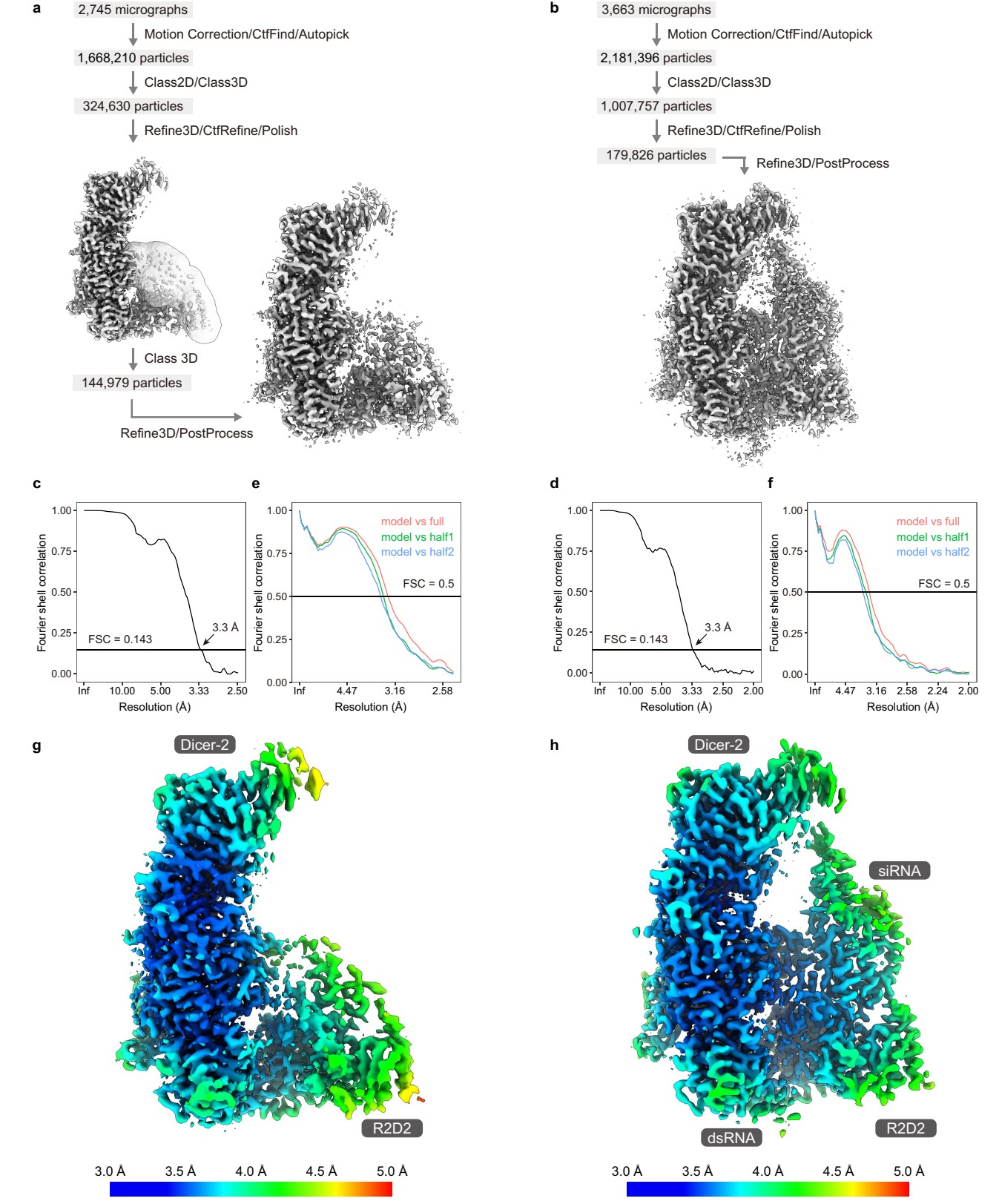

**Extended Data Fig. 1 | Cryo-EM analysis. (a, b)** Single-particle cryo-EM image processing workflows for Dicer-2–R2D2 (**a**) and Dicer-2–R2D2–siRNA (**b**). (**c, d**) Fourier shell correlation curves for the 3D reconstructions of Dicer-2–R2D2 (**c**) and Dicer-2–R2D2–siRNA (**d**). (**e, f**) Fourier shell correlation curves calculated between the refined models and the density maps for Dicer-2–R2D2 (**e**) and Dicer-2–R2D2–siRNA (**f**). (**g, h**) Local-resolution density maps of Dicer-2–R2D2 (**g**) and Dicer-2–R2D2–siRNA (**h**).

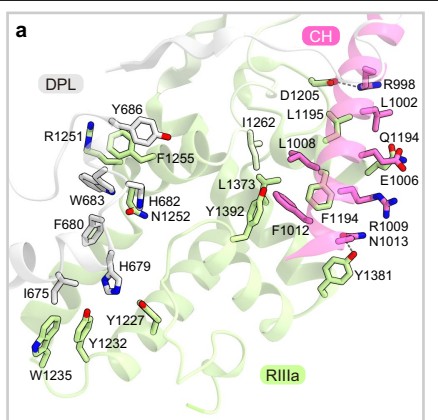

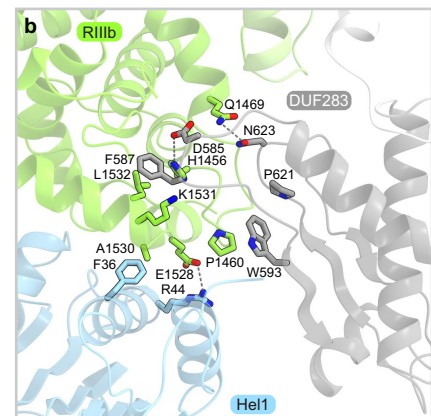

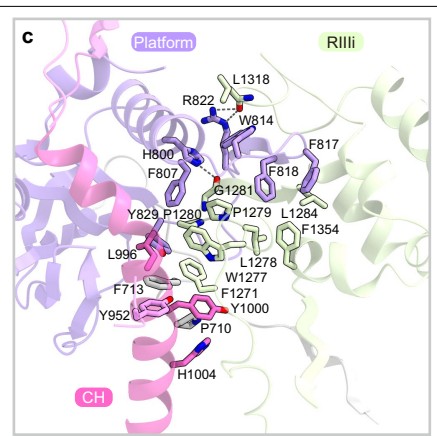

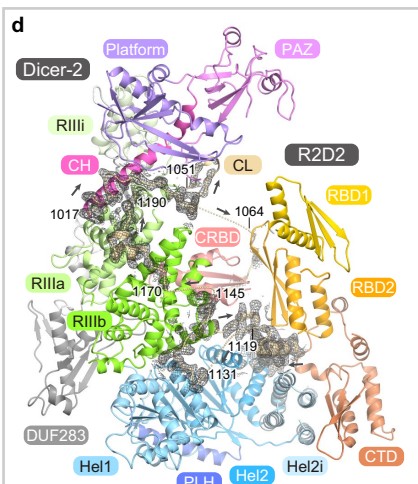

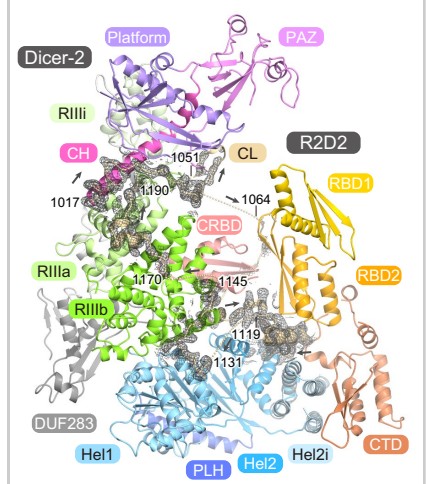

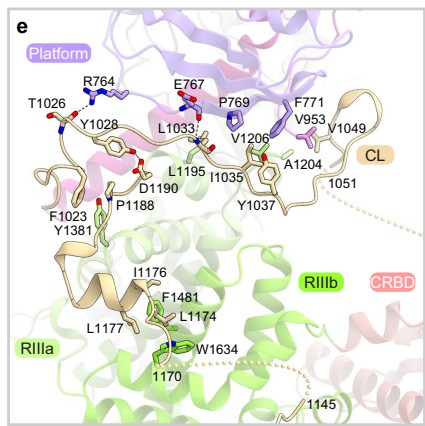

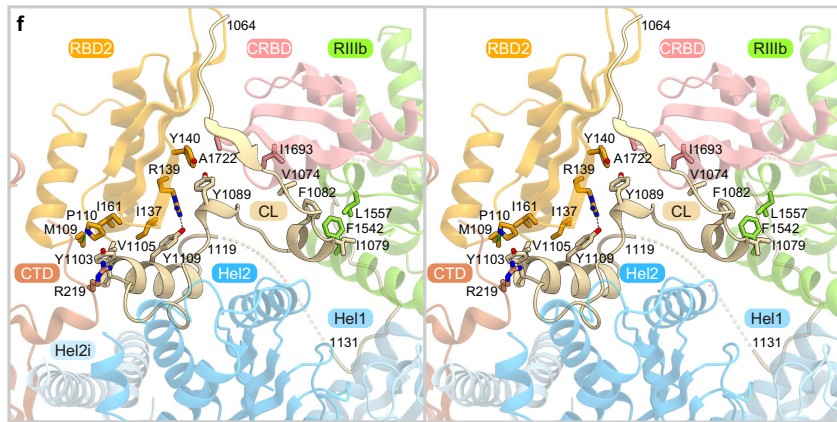

**Extended Data Fig. 2 | Domain interfaces.** (**a**) DPL–CH–RIIIa interface.
(**b**) Hel1–DUF283–RIIIb interface. (**c**) Platform–CL–RIIIi interface. (**d**) Density
map for the central linker (contoured at 9σ) (stereo view). (**e**) Platform–CL–RIIIa
interface. (**f**) CL–RIIIb–CRBD–RBD2–CTD interface (stereo view). Y1089 in the
central linker and A1722 in CRBD form hydrophobic interactions with Y140 in

R2D2 RBD2. Y1103 in Dicer-2 forms stacking interactions with P110 and R219 in
R2D2. V1105 of Dicer-2 hydrophobically interacts with M109, I137, and I161 of
R2D2. Y1109 of Dicer-2 hydrogen bonds with R139 in R2D2. H224 and M292 in
R2D2 CTD are accommodated within hydrophobic pockets formed by R334/
H335/V338 and F245/M252/L321/M322/T325 in Dicer-2 Hel2i, respectively.

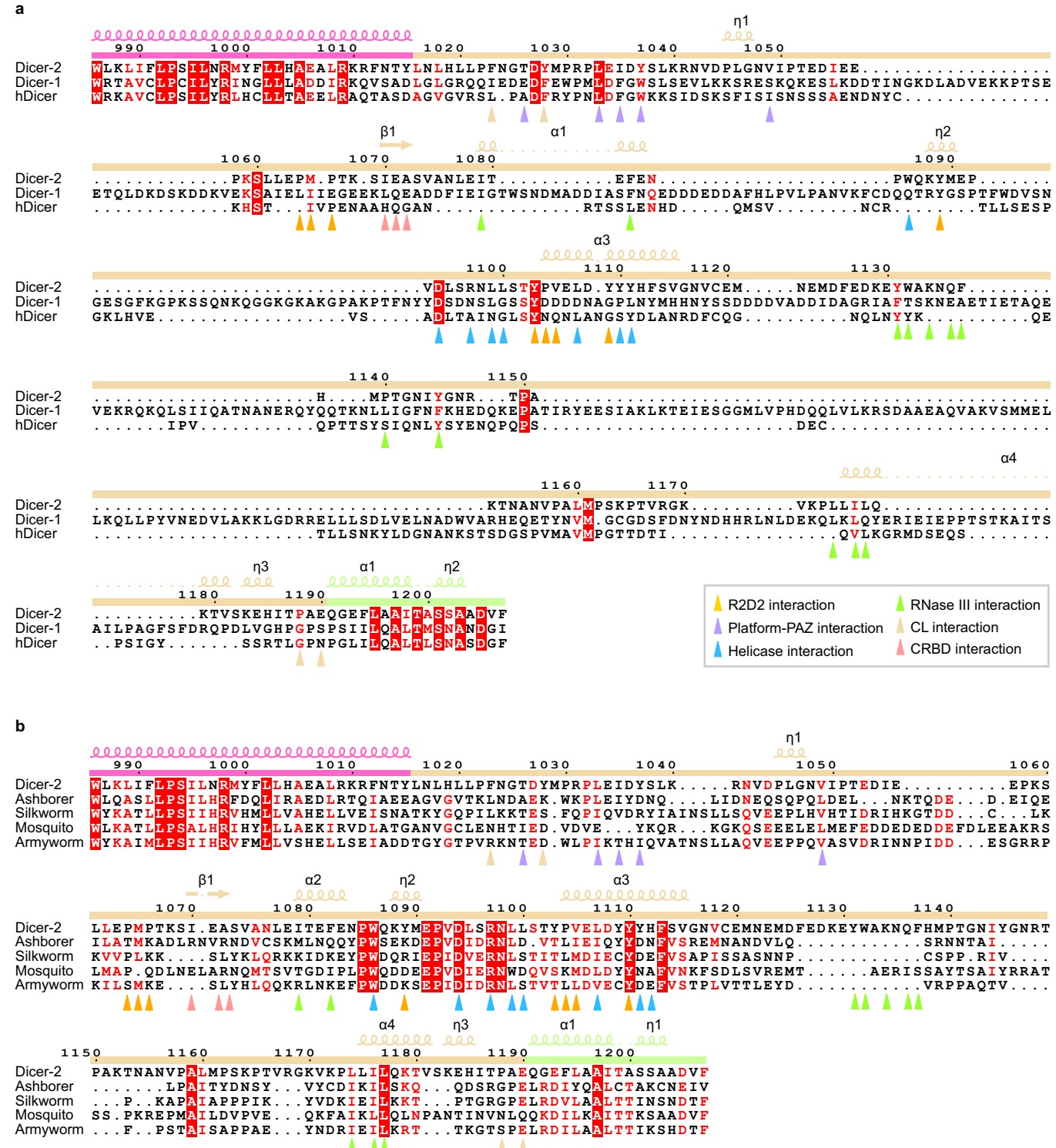

**Extended Data Fig. 3 | Multiple sequence alignments of the central linker regions.** (**a**, **b**) Multiple sequence alignments of the central linker regions in Dicer-2, Dicer-1, and human Dicer (**a**), and the Dicer-2 orthologs from insect species (**b**). The secondary structure of Dicer-2 is indicated above the sequences. The connector helix, the central linker, and the RNase III domain are highlighted in pink, beige, and light green, respectively. The key residues are marked by triangles.

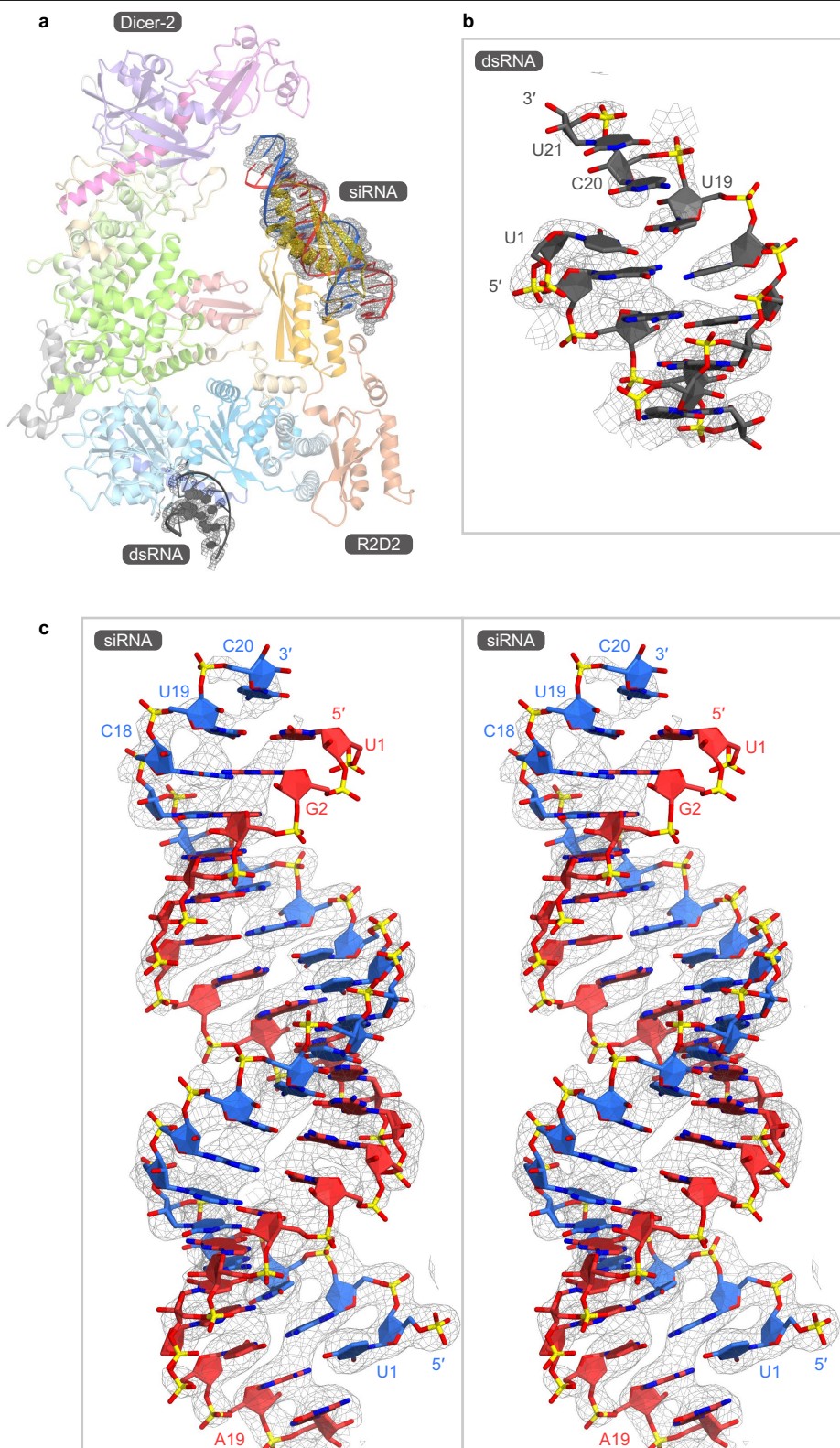

**Extended Data Fig. 4 | Density maps for the bound RNA.** (**a–c**) Density maps for dsRNA and siRNA (contoured at 4σ) (**a**), dsRNA (contoured at 4σ) (**b**), and siRNA (contoured at 6σ) (**c**) (stereo view).

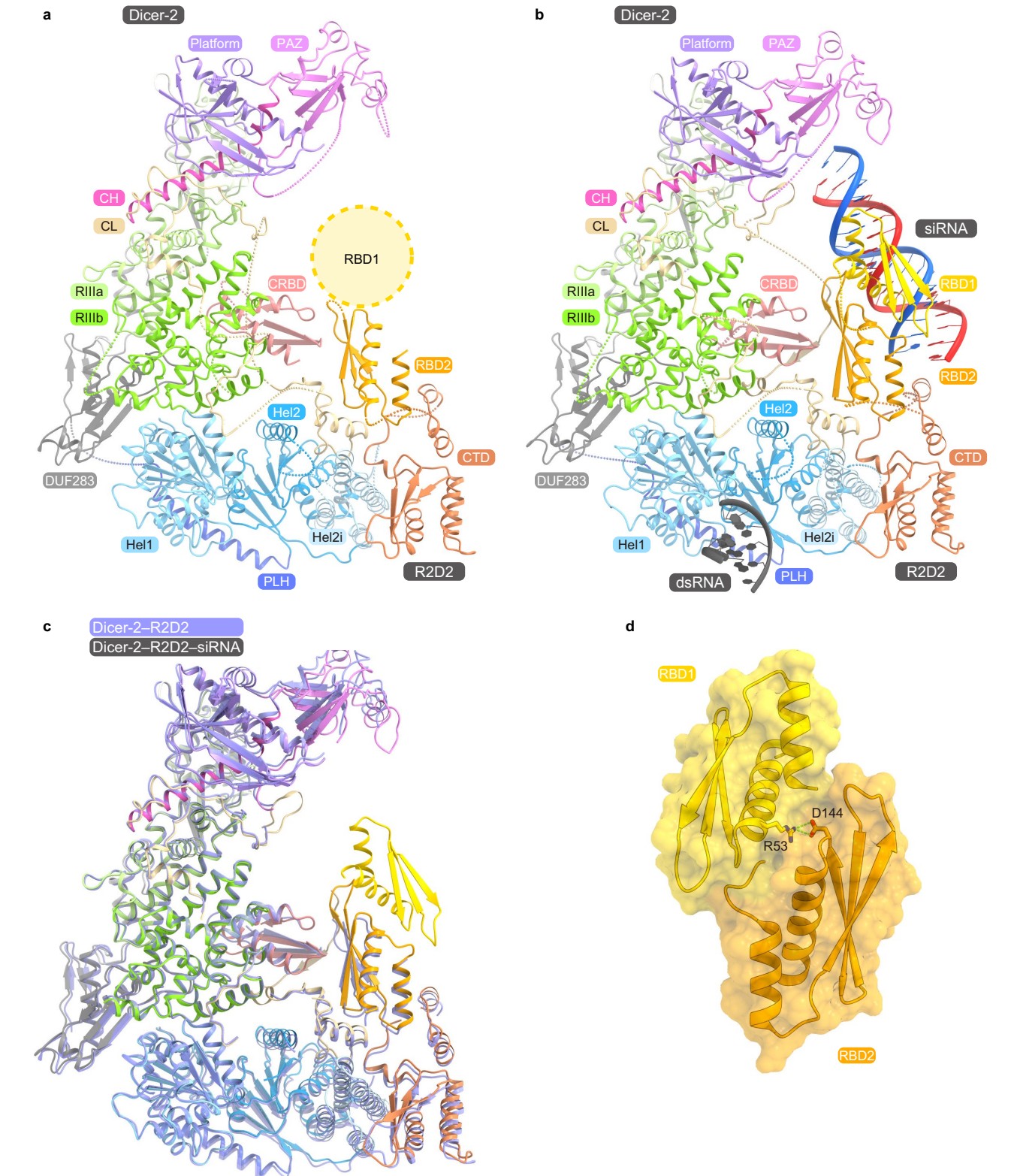

**Extended Data Fig. 5 | Structural comparison between the Dicer-2–R2D2 and Dicer-2–R2D2–siRNA complexes.** (**a, b**) Cryo-EM structures of the Dicer-2–R2D2 heterodimer (**a**) and the Dicer-2–R2D2–siRNA complex (**b**). (**c**) Superimposition between Dicer-2–R2D2 (blue) and Dicer-2–R2D2–siRNA (colored as in **b**). The two structures were superimposed based on their RNase III domains. (**d**) Interaction between RBD1 and RBD2 of R2D2 in the Dicer-2–R2D2–siRNA complex. R53 of RBD1 forms a salt bridge with D144 of RBD2, stabilizing the RBD1–RBD2 interface.

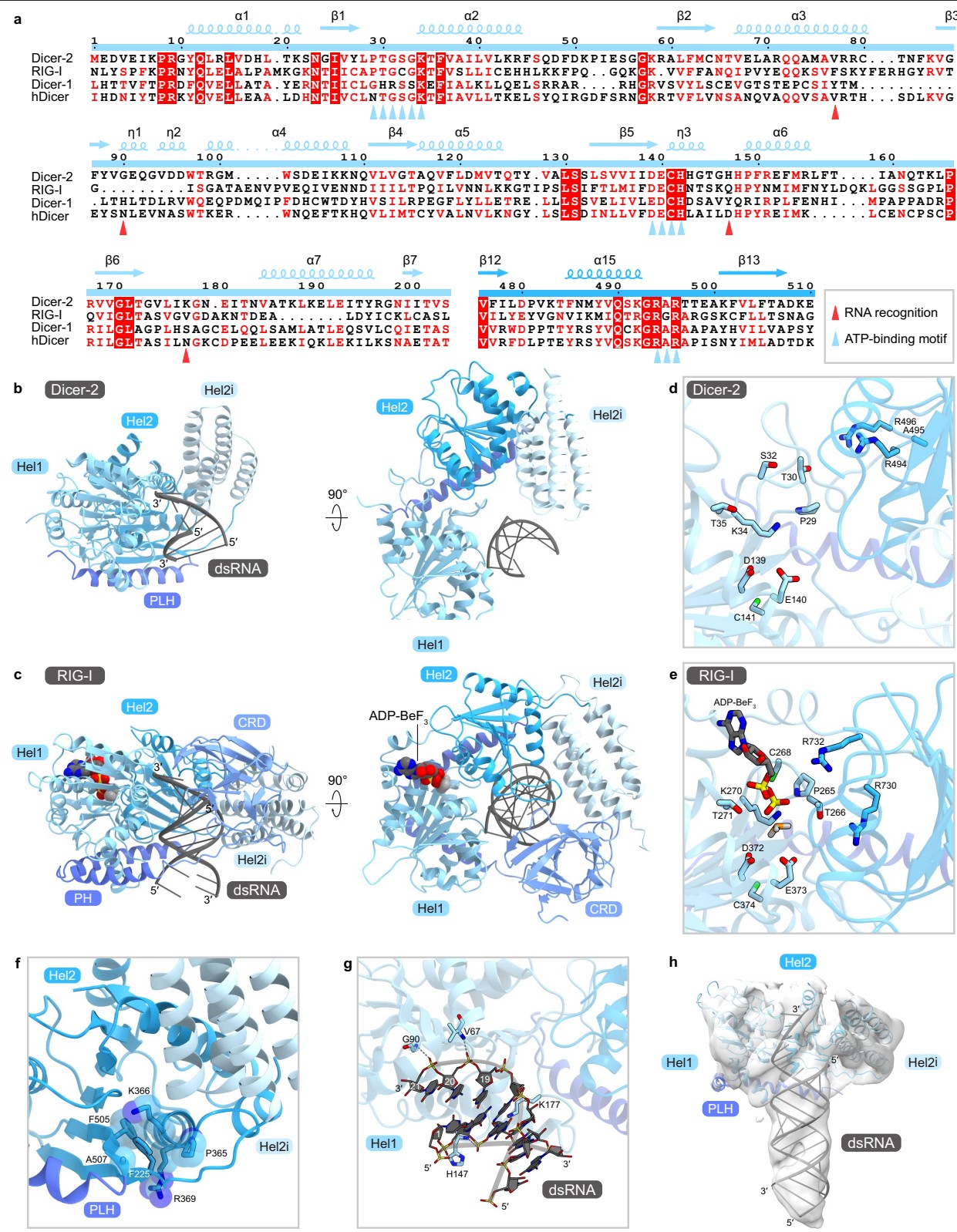

**Extended Data Fig. 6 | Helicase domain of Dicer-2. (a)** Multiple sequence alignment of Dicer-2, RIG-I, Dicer-1, and human Dicer. The secondary structure of Dicer-2 is indicated above the sequences. The Hel1 and Hel2 domains are highlighted in light blue and blue, respectively. The key residues are marked by triangles. The figure was prepared using Clustal Omega (http://www.ebi.ac.uk/Tools/msa/clustalo) and ESPript3 (http://espript.ibcp.fr/ESPript/ESPript). **(b, c)** Helicase domains of Dicer-2 **(b)** and RIG-I (PDB: 3TMI) **(c)**. CRD, carboxy-terminal regulatory domain. **(d, e)** ATP-binding sites of Dicer-2 **(d)** and RIG-I (PDB: 3TMI) **(e)**. **(f)** Hydrophobic core in the Dicer-2 helicase domain. **(g)** Recognition of dsRNA by the Dicer-2 helicase domain. **(h)** Cryo-EM reconstitution of the Dicer-2 helicase domain bound to a blunt-end dsRNA (PDB: 6BU9). A RIG-I-based homology model was fitted to the 6.8-Å resolution density map as a rigid body.

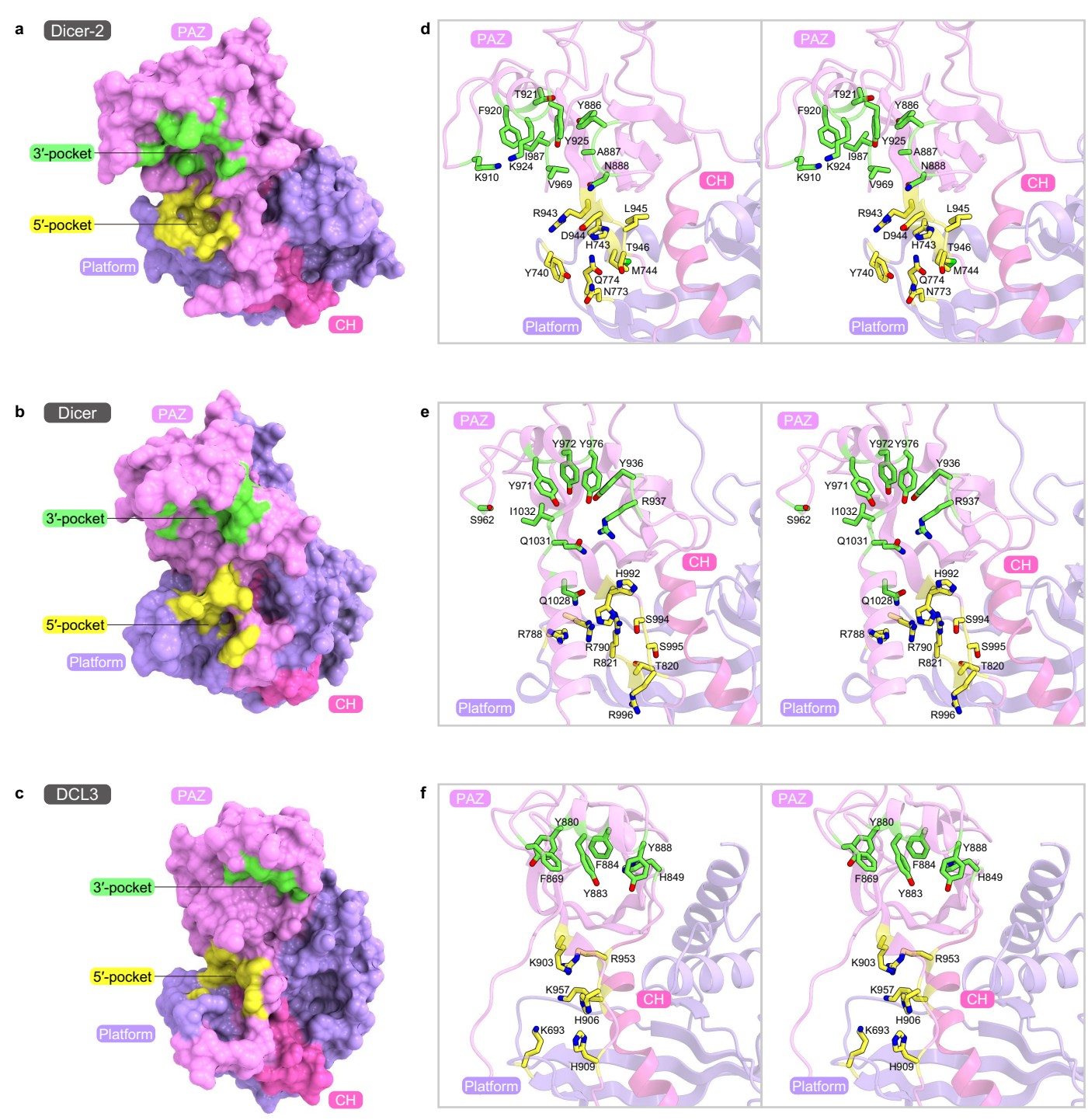

**Extended Data Fig. 7 | Structural comparison between the Platform-PAZ domains of Dicer-2, human Dicer, and plant DCL3.** (**a**–**c**) Platform-PAZ domains of Dicer-2 (**a**), human Dicer (PDB: 5ZAL) (**b**), and DCL3 (PDB: 7VG3) (**c**). (**d**–**f**) 5′- and 3′-pockets of Dicer-2 (**d**), human Dicer (PDB: 5ZAL) (**e**), and DCL3 (PDB: 7VG3) (**f**) (stereo view).

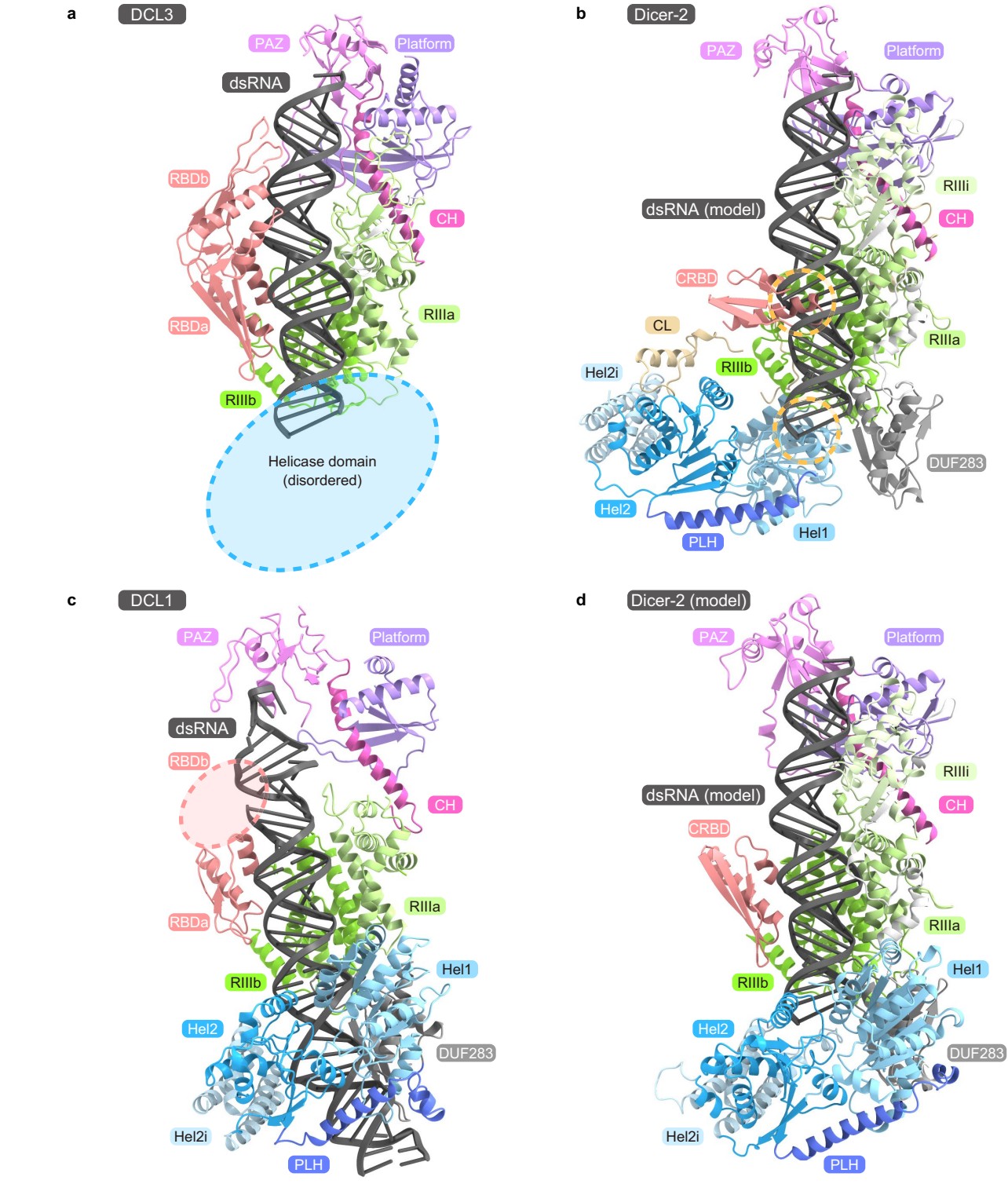

**Extended Data Fig. 8 | Structural comparison between Dicer-2, DCL1, and DCL3. (a)** Structure of the DCL3–dsRNA complex (PDB: 7VG3). The amino-terminal helicase domain is disordered in the structure. **(b)** Model of the Dicer-2–dsRNA complex. The DCL3–dsRNA structure was superimposed onto the Dicer-2 structure, based on their RNase III domains, and DCL3 was then omitted. Predicted steric clashes are indicated by dashed circles. **(c)** Structure of the DCL1–dsRNA (pri-miRNA) complex (PDB: 7ELD). The carboxy-terminal RBDb domain is disordered in the structure. **(d)** Model of the Dicer-2–dsRNA complex. The Dicer-2 structure was predicted by AlphaFold2, and the dsRNA and CRBD were modeled based on the DCL3–dsRNA structure. Unstructured regions in the RIIIb and CL domains are omitted for clarity. In the model, the helicase domain interacts with the DUF283 domain and recognizes the dsRNA substrate, as in the DCL1–dsRNA structure.

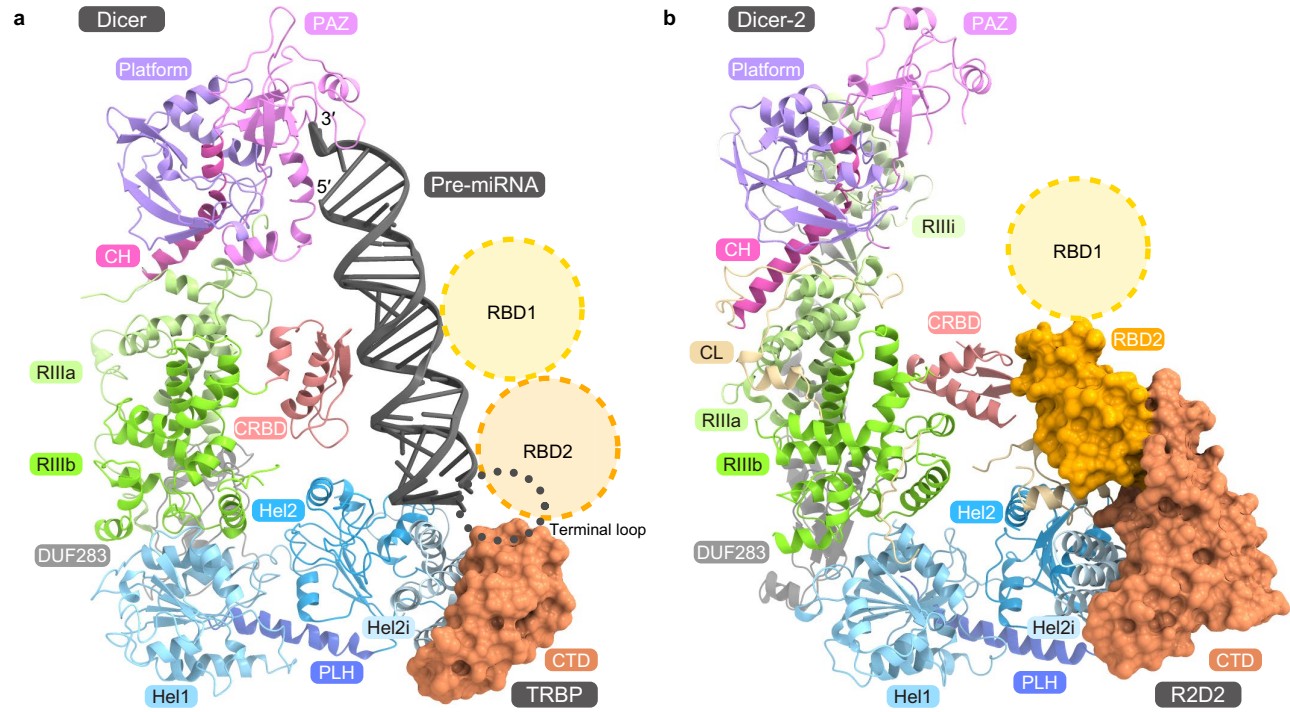

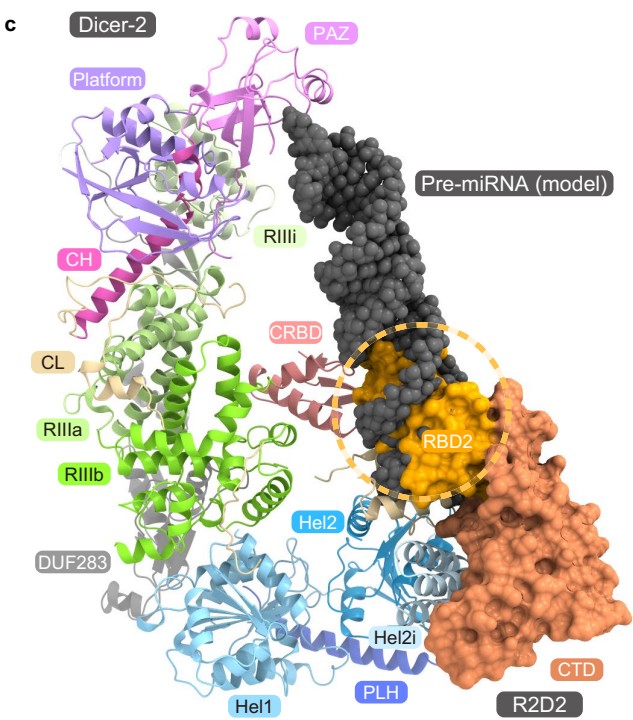

**Extended Data Fig. 9** | See next page for caption.

**Extended Data Fig. 9 | Structural comparison between Dicer-2 and human Dicer.** (**a**) Structure of the human Dicer–TRBP heterodimer bound to a pre-miRNA (pre-*let-7*) (PDB: 5ZAL). TRBP is shown as a surface model. RBD1 and RBD2 of TRBP are disordered in the structure and indicated by dashed circles. The human Dicer–TRBP–pre-*let-7* structure represents the pre-miRNA substrate recognition state, in which the 2-nt 3′-overhanging end and the terminal loop of pre-*let-7* are anchored by the Platform-PAZ domain and the interface between human Dicer Hel2i and TRBP CTD, respectively. (**b**) Structure of the Dicer-2–R2D2 heterodimer. R2D2 is shown as a surface model. RBD1 of R2D2 is disordered in the structure and indicated by a dashed circle. (**c**) Possible steric clash between R2D2 and a pre-miRNA. Dicer-2 initially recognizes a long dsRNA substrate using only the helicase domain, whereas human Dicer recognizes a pre-miRNA substrate using both the PAZ and helicase domains. Based on the hypothesis that, like human Dicer, Dicer-2 recognizes a pre-miRNA using both the PAZ and helicase domains, we modeled a pre-miRNA onto the Dicer-2–R2D2 structure, as follows. The human Dicer–TRBP–pre-miRNA (pre-*let-7*) structure was superimposed onto the Dicer-2–R2D2 structure based on their PAZ domains, and human Dicer and TRBP were then omitted. While Dicer-2 and human Dicer commonly associate with their partner proteins via their Hel2i–CTD interfaces, conformational differences are present in RBD1 and RBD2 of the partner proteins. RBD2 of R2D2 is ordered and interacts with the central linker and CRBD in the Dicer-2–R2D2 structure (RBD1 is disordered). In contrast, RBD1 and RBD2 of TRBP are disordered and may loosely interact with the pre-miRNA in the human Dicer–TRBP–pre-miRNA structure. A structural comparison of Dicer-2–R2D2 with human Dicer–TRBP–pre-miRNA suggested that R2D2 could sterically clash with a modeled pre-miRNA, explaining why R2D2 inhibits promiscuous pre-miRNA processing by Dicer-2.

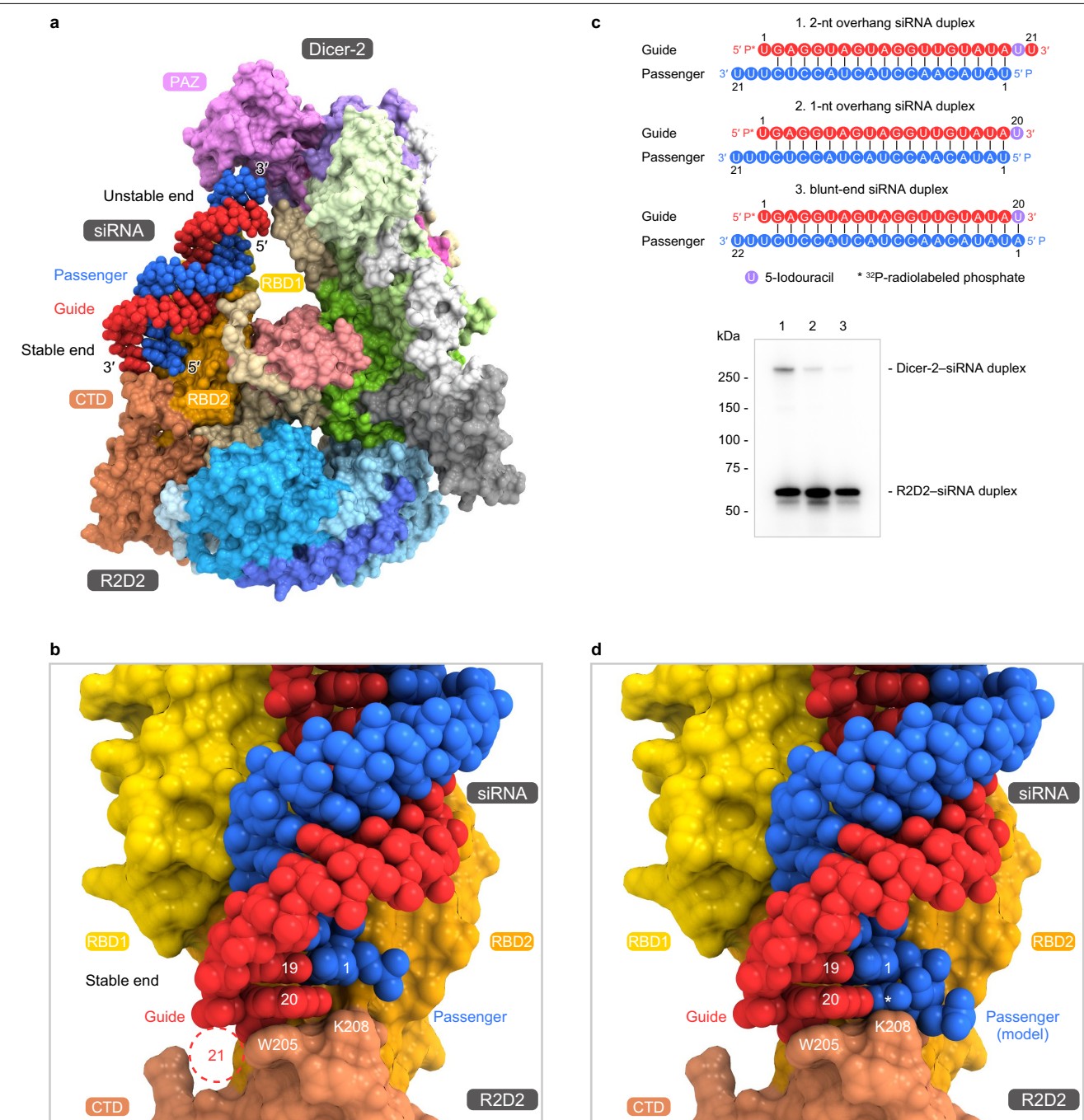

**Extended Data Fig. 10 | Recognition of the siRNA duplex by the Dicer-2–R2D2 heterodimer.** (**a**) Asymmetric recognition of the siRNA duplex by the Dicer-2–R2D2 heterodimer. (**b**) Recognition of the 1-nt 3′-overhang (G20) of the siRNA duplex by R2D2. U21 is disordered in the structure, and a possible location of U21 is indicated by a dashed circle. (**c**) Photocrosslinking experiments. Dicer-2 and R2D2 were incubated with 5′-radiolabeled siRNA containing 5-iodouracil at position 20 and a 0–2-nt 3′-overhang at the stable end. The reaction mixture was analysed by SDS-PAGE, and crosslinked proteins were detected using phosphorimaging (*n* = 3 independent experiments). (**d**) Possible interaction between R2D2 and a blunt-end siRNA. A modeled nucleotide complementary to G20 in the guide strand is indicated by an asterisk.

# Reporting Summary

## Statistics

For all statistical analyses, confirm that the following items are present in the figure legend, table legend, main text, or Methods section.

| n/a | Confirmed | |
|---|---|---|
| ☐ | ☒ | The exact sample size ($n$) for each experimental group/condition, given as a discrete number and unit of measurement |
| ☐ | ☒ | A statement on whether measurements were taken from distinct samples or whether the same sample was measured repeatedly |
| ☒ | ☐ | The statistical test(s) used AND whether they are one- or two-sided *Only common tests should be described solely by name; describe more complex techniques in the Methods section.* |
| ☒ | ☐ | A description of all covariates tested |
| ☒ | ☐ | A description of any assumptions or corrections, such as tests of normality and adjustment for multiple comparisons |
| ☒ | ☐ | A full description of the statistical parameters including central tendency (e.g. means) or other basic estimates (e.g. regression coefficient) AND variation (e.g. standard deviation) or associated estimates of uncertainty (e.g. confidence intervals) |
| ☒ | ☐ | For null hypothesis testing, the test statistic (e.g. $F$, $t$, $r$) with confidence intervals, effect sizes, degrees of freedom and $P$ value noted *Give P values as exact values whenever suitable.* |
| ☒ | ☐ | For Bayesian analysis, information on the choice of priors and Markov chain Monte Carlo settings |
| ☒ | ☐ | For hierarchical and complex designs, identification of the appropriate level for tests and full reporting of outcomes |
| ☒ | ☐ | Estimates of effect sizes (e.g. Cohen's $d$, Pearson's $r$), indicating how they were calculated |

*Our web collection on statistics for biologists contains articles on many of the points above.*

## Software and code

Policy information about availability of computer code

| Data collection | Serial EM (version 3.7.10) |
|---|---|
| Data analysis | RELION (version 3.0 and version 3.1), Servalcat (version 0.2.0), COOT (version 0.9), UCSF ChimeraX (version 1.1.1), CueMol2 (http://www.cuemol.org/ version 2.2.3.443) |

For manuscripts utilizing custom algorithms or software that are central to the research but not yet described in published literature, software must be made available to editors and reviewers. We strongly encourage code deposition in a community repository (e.g. GitHub). See the Nature Portfolio guidelines for submitting code & software for further information.

## Data

Policy information about availability of data

All manuscripts must include a data availability statement. This statement should provide the following information, where applicable:

- Accession codes, unique identifiers, or web links for publicly available datasets
- A description of any restrictions on data availability
- For clinical datasets or third party data, please ensure that the statement adheres to our policy

The cryo-EM density maps and atomic coordinates have been deposited in the Electron Microscopy Data Bank. The accession codes for the maps are EMD-31741 (Dicer-2–R2D2) and EMD-31742 (Dicer-2–R2D2–siRNA). The accession codes for the coordinates are 7V6B (Dicer-2–R2D2) and 7V6C (Dicer-2–R2D2–siRNA).

# Field-specific reporting

Please select the one below that is the best fit for your research. If you are not sure, read the appropriate sections before making your selection.

☒ Life sciences  ☐ Behavioural & social sciences  ☐ Ecological, evolutionary & environmental sciences

For a reference copy of the document with all sections, see nature.com/documents/nr-reporting-summary-flat.pdf

# Life sciences study design

All studies must disclose on these points even when the disclosure is negative.

| | |
|---|---|
| Sample size | For cryo-EM analyses, sample sizes were determined by the availability of microscope time and the number of particles on electron microscopy grids enough to obtain a structure at the reported resolution. For biochemical analysis, sample size were determined based on the previous reports of this type of study (Tomari et al., 2004) and the reproducibility of results across independent experiments. |
| Data exclusions | For cryo-EM analyses, particles that did not contribute to improving map quality were excluded following the standard classification procedures in RELION. This is standard practice for structure determination by cryo-EM. For biochemical analyses, no data was excluded. |
| Replication | For cryo-EM analyses, related experiments including purification, and SDS-PAGE were reproduced at least two times and structure determination was completed once based on the previous reports of this type of study. For biochemical analyses, all measurements were repeated at least three times. All attempts at replication were successful. |
| Randomization | For cryo-EM analyses, particles were randomly assigned to half-maps for resolution determination following the standard procedures in RELION. For biochemical analyses, randomization was not performed because this study does not involve animal experiments. |
| Blinding | Blinding is not applicable since this is an exploratory study and blinding is impossible or unlikely to affect the results or interpretation of the results. |

# Reporting for specific materials, systems and methods

We require information from authors about some types of materials, experimental systems and methods used in many studies. Here, indicate whether each material, system or method listed is relevant to your study. If you are not sure if a list item applies to your research, read the appropriate section before selecting a response.

### Materials & experimental systems

| n/a | Involved in the study |
|---|---|
| ☐ | ☒ Antibodies |
| ☐ | ☒ Eukaryotic cell lines |
| ☒ | ☐ Palaeontology and archaeology |
| ☒ | ☐ Animals and other organisms |
| ☒ | ☐ Human research participants |
| ☒ | ☐ Clinical data |
| ☒ | ☐ Dual use research of concern |

### Methods

| n/a | Involved in the study |
|---|---|
| ☒ | ☐ ChIP-seq |
| ☒ | ☐ Flow cytometry |
| ☒ | ☐ MRI-based neuroimaging |

# Antibodies

| | |
|---|---|
| Antibodies used | anti-FLAG-antibody (Sigma-Aldrich F3165, 1/2000)<br>anti-Dicer-2-antibody (Abcam ab4732, 1/1000)<br>Goat anti-mouse IgG (H+L)/(H&L) antibody, HRP conjugate (proteintech SA00001-1, 1/4000)<br>Goat anti-rabbit IgG (H+L)/(H&L) antibody, HRP conjugate (proteintech SA00001-2, 1/4000) |
| Validation | anti-FLAG-antibody (Sigma-Aldrich F3165) https://www.sigmaaldrich.com/specification-sheets/120/274/F3165-BULK.pdf<br>anti-Dicer-2-antibody (Abcam ab4732) https://www.abcam.co.jp/dcr-2--dicer-2-antibody-ab4732.html<br>Goat anti-mouse IgG (H+L)/(H&L) antibody, HRP conjugate https://www.ptglab.co.jp/products/HRP-conjugated-Affinipure-Goat-Anti-Mouse-IgG-H-L-secondary-antibody.htm<br>Goat anti-rabbit IgG (H+L)/(H&L) antibody, HRP conjugate https://www.ptglab.co.jp/products/HRP-conjugated-Affinipure-Goat-Anti-Rabbit-IgG-H-L-secondary-antibody.htm |

# Eukaryotic cell lines

Policy information about cell lines

| | |
|---|---|
| Cell line source(s) | Sf9 (ATCC, Cat.#CRL-1711) |
| Authentication | sf9 cells were purchased from ATCC Cell lines and no further authentication was performed. |
| Mycoplasma contamination | sf9 cells were not tested for mycoplasma contamination. |
| Commonly misidentified lines (See ICLAC register) | sf9 cells are not misidentified cell lines. |

