## [Peer Review File · Nature]

Manuscript Title: Structure of the Dicer-2–R2D2 heterodimer bound to a small RNA duplex

Reviewer Comments & Author Rebuttals

Reviewer Reports on the Initial Version:

Referees' comments:

Referee #1 (Remarks to the Author):

Yamaguchi and co-workers present the structure of the Dcr2-R2D2 heterodimer with and without bound siRNA duplexes. Many in the RNAi field, including myself, have waited many years to see this structure and the results do not disappoint. The asymmetry rule for siRNA guide strand selection was established nearly two decades ago. While several asymmetry sensors have been proposed, the best characterized is the *Drosophila* Dcr2-R2D2 heterodimer. This complex is also essential for siRNA production in flies and has been an instrumental part of mechanistic studies of RNAi. Yamaguchi et al., now reveal the structure of the heterodimer bound to an siRNA duplex at atomic resolution. The structure provides the most accurate and complete structure of any eukaryotic Dicer protein, the atomic details of asymmetry sensing by R2D2, and insights into dsRNA recognition and processing by the Dicer helicase. The study was a joy to read and, in my opinion, will make a fantastic contribution.

I have the following questions, primarily surrounding the details of asymmetry sensing:

1) Line 223: “The unstable end interacts with neither Dicer-2 nor R2D2...”. This is surprising considering data in Fig. 5G and previous results (PMID: 15550672) showing the unstable siRNA end efficiently crosslinks to Dcr-2. If asymmetry sensing is entirely through R2D2, as the structure would suggest, why is crosslinking to Dcr2 unaffected by the W205A mutation? Is it possible that Dcr2 does engage the unstable siRNA end, but this conformation of the complex was not captured in the single particle analysis?

2) Fig. 5E indicates stability at the very end of the siRNA duplex is detected by stacking of guide strand nucleobase-20 with W205, and passenger nucleobase-1 with the aliphatic portion of K98. Fig 5G. shows guide crosslinking to R2D2 is lost in the W205A mutant. Have the authors determined how the W205A mutation impacts siRNA affinity? Have the authors examined crosslinking in the K98A mutant?

3) The structure shown Fig. 5E indicates RD2D actually only recognizes one nucleotide in the siRNA 3' overhang (as nucleotide 21 is disordered). Does biochemistry support this mechanism? i.e. would a shortened siRNA, with a single nucleotide 3' overhang, crosslink equally well or is a 2 nucleotide overhang needed?

4) Line 248: “the W205A mutation and the CTD α 2 deletion abolished the crosslinking of g* to

R2D2..... indicating that the siRNA stable end does not form a stacking interaction with these R2D2 mutants.” This sentence is confusing....how do the authors know crosslinking correlates with stacking in the CTD α 2 deletion? For the W205A mutant, how could stacking interactions be formed in the first place?

5) Fig. 3E: why are the two strands designated “guide” and “passenger”? Is this a typo or are the authors suggesting that orientation bound to the helicase can determine which strand will be loaded as guide? I ask because such a notion is difficult to reconcile with the model for siRNA release and rebinding after dicing. Also, considering the limited quality of the density in Extended Data Fig. 3B, I am wondering how the authors can know which end of the siRNA duplex is bound.

6) The mechanism for Ago-loading in Supplementary Movie 2 is intriguing because it predicts the relative positions of Dicer2 and Ago2. Sasaki and Shimizu identified an Ago-binding motif in the RNaseIIIa domain of human dicer many years ago (PMID: 17482383). Is the equivalent region of DmDcr2 near the surface predicted to interact with Ago2?

7) Line 147: “...that central linker (residues 1081–1115) interacts with...”. Residues is misspelt, but more importantly, the residue numbers seem incorrect.

8) Supplementary Movie 1: the central linker (CL) is mislabeled “CH”.

Referee #2 (Remarks to the Author):

This study provides cryo-EM structures of fly Dicer-2-R2D2 heterodimer and Dicer-2-R2D2-siRNA complex at high resolutions. These structures contain previously unidentified regions including a linker between the Platform-PAZ and RIII domains (“central linker”) and a large α -helical domain within the RIIIa domain (“RIIIi”). These structures of these domains are difficult to predict using AlphaFold2 and turn out to form extensive contacts with other domains of Dicer-2 and R2D2. The authors captured two different binding modes of siRNA substrates, presumably indicating the initial recognition state and pre-loading state, respectively. In particular, the pre-loading state structure provides atomic-level insights into how Dicer-2 and R2D2 sense siRNA thermodynamic asymmetry, for the first time.

Although a low-resolution structure (7.1 Å) of fly Dicer-2-R2D2 complex with dsRNA bound to the helicase domain (which represents the pre-Dicing state) was previously reported (Sinha et al., 2018), the current study substantially advances our understanding of small RNA biology by solving a high-resolution (3.3 Å) structure in the pre-loading state.

Major comments:

1. The authors show, in the structure and with photo-crosslinking experiments, that R2D2 with specific amino acids interacts with the central region of the siRNA duplex and forms a stacking interaction with the base of g20. However, the major difference between the stable and unstable ends of siRNA duplex lies in the Watson-Crick base-pairing at the end of the stem (g19) between the

two strands. There is no clear structural explanation of how R2D2 senses such base-pairing difference. Therefore, building structural models with the unstable end positioned in the pocket formed by the dsRBD2 and CTD of R2D2 might help get insights to explain the asymmetry mechanism much clearer.

2. They argued that W205 and CTD $\alpha 2$ of R2D2 are critical for the siRNA thermodynamic asymmetry based on the photocrosslinking assay (Figure 5G). However, in contrast to the deletion of CTD $\alpha 2$, the W205A mutation didn't show a significant increase of g^* binding. Is it possible that the W-to-A mutation just abolishes the crosslinking not the asymmetric binding? If it is difficult to exclude this possibility, please focus on the importance of CTD $\alpha 2$.

3. The authors suggested that the central linker (1081-1115 aa) has amino acid residues specifically interacting with R2D2 (Figure 2A). These amino acids include Y1089, Y1103, V1105, and V1109. However, the pulldown experiments show that FLAG-tagged R2D2 has no problem pulling down Dicer-2 mutant ($\Delta 1082-1115$) that lacks all four amino acid residues above. Although a smaller amount of Dicer-2 mutant was pulled down compared to when wild-type Dicer-2 was expressed, but this is likely because of the low expression levels of Dicer-2 mutant (as observed when pulled down with a His tag). It would be great if the authors can elaborate more on this, whether the interaction would be, for example, meaningful in the context of processing substrates.

4. As mentioned in the paper, "Dicer-2, but not Dicer-2-R2D2, can process pre-miRNAs, indicating that R2D2 acts in determining the specificity of Dicer-2 for the dsRNA substrates" (there is no citation, by the way). According to the model of siRNA production by Dicer-2-R2D2 proposed in Figure 7, Dicer-2 recognizes a long dsRNA substrate via the helicase domain. In the structure, however, R2D2 is isolated from where initial substrate recognition is made and it is difficult to imagine how R2D2 can block pre-miRNA processing, especially when the helicase domain recognizes not only blunt ends but also 2-nt 3'-overhang (as observed in the structure). Although the authors illustrated how R2D2 might sterically clash with a pre-miRNA that enters from the center, it would be great to know how the authors perceive this scenario.

Minor comments:

1. Please mark the CTD $\alpha 2$ in Figure 5B in addition to Figure 5E, which would help the understanding.

2. Typos

- Line 103, reference 37 → reference 30 (and its reference style also differs from the others)
- Lines 111 and 115, Figures 2X → Extended Data Figures 2X (?)
- Line 146: residusses

Referee #3 (Remarks to the Author):

Yamaguchi et al.

Structure of the Dicer-R2D2 heterodimer bound to small RNA duplex

In this manuscript, the authors have solved the cryoEM structures of the *Drosophila* Dicer2-R2D2 complex and the Dicer2-R2D2-ds-siRNA complex. The resolution is 3.3 Å and allows for building de novo atomic models. The structures are in a post-cleavage conformation where the siRNA is stably bound by R2D2. This is expected since a short RNA has been included and cleavage should not occur. With their structures, the authors confirm and validate several mechanistic aspects that have been suggested before based on biochemical experiments. In addition, there are also several novel features. Comparing the two complexes, the authors revealed that the dsRbd1 of R2D2 becomes ordered when it binds to the central region of the siRNA duplex. The interaction surface of Dicer-2 and R2D2 is also further refined on an atomic level. Comparison of the helicase domains of Dicer-2 and RIG-I also allows for conclusions on dsRNA recognition of this domain although the bound RNA is not well resolved and thus only modeled. The authors also compare the active site and the 5' and 3' binding pockets with a human cryoEM structure, that is in a different conformation. The authors present several mechanistic assumptions based on the two different conformations. Finally, the authors analyzed the strand selection by R2D2 and further confirm that R2D2 anchors the siRNA in an orientation in which the less stable end is exposed to the solvent and the more stable end is anchored on R2D2.

This is a very clear, well written manuscript presenting an important structure of the Dicer-2-R2D2-siRNA complex that deserves publication in such a top journal. The structure allows for a better understanding of the mechanisms underlying strand selection and subsequent siRNA loading into Ago. However, many of the observed features confirm known models and several aspects are solely based on modeling and assumptions. Listed below are a number of points/issues that should be clarified or discussed.

1. On page 5, lines 128-130: the authors report that the density of the siRNA bound to the helicase domain was relatively unclear suggesting flexibility. Thus, they modeled the unstable end of siRNA-1 according to the shapes of the densities. This needs to be explained better. If the densities are so clear that the unstable end can be modeled, then it is not clear why the siRNA cannot be better defined. Otherwise the modeling is rather random and might not be a very strong basis for the mechanistic conclusions that are drawn later when comparing it with the RIG-I structure. Particularly, the conclusions on blunt end and overhang binding seems to be rather speculative. Extended data Figure 5D is not mentioned in the text.
2. Page 5, bottom: R2D2 dsRBD1 is disordered in the siRNA unbound state. What is the evidence that it is disordered? If the correctly folded domain would be highly mobile and flexible in the unbound state, it would also not be resolved in the structure. Or are there other data suggesting that it is disordered?
3. On Page 7: the authors model long dsRNA into their structure and find that it clashes at various

positions. This leads to the conclusion that there must be mechanistically important rearrangements during siRNA production. Although this is likely true, the conclusion from the data is somewhat far-fetched based on a comparison of hDicer and *Drosophila* Dicer-2. I think the final statement that "...our structure provides mechanistic insights into Dicer-2 catalyzed high-fidelity siRNA production" is not justified and should be rephrased and toned down.

4. Page 8, last sentence of second paragraph: "A modeled blunt siRNA sterically clashes...". This is unclear. First, 'data not shown' should be added. Second, isn't a blunt end siRNA duplex also shown in 5E? And why would a blunt end clash while overhangs not? This is confusing and should be described clearer.

5. Figure 5F: the authors state several times that the less stable end is exposed to the solvent. On the other hand, both ends can be cross linked to the complex. This should be consistent and described clearer.

6. One of the most intriguing question in this context is how the more stable and the less stable ends are sensed. I understand that there is some kind of dissociation and association and during this step the correct orientation is achieved. But how is it sensed? This is not really addressed in this study. The conclusions made here seem to be mainly confirmative of already proposed models. Are there specific residues that would prefer G-C pairs over A-U? Or are there sterical features that would require a stable structure and less 'breathing' of the end? Maybe the analyses could be extended into this direction.

7. Reference to extended data Figures 1 and 2 are not accurate in the first two paragraphs of the results section. This should be revisited and corrected.

Author Rebuttals to Initial Comments:

We appreciate the referees' suggestions and concerns, which have significantly improved our manuscript. According to their comments, we performed additional crosslinking experiments and revised the manuscript. In addition, we compared the structure of Dicer-2 with those of the plant Dicer enzymes (DCL1 and DCL3), which were recently reported. Furthermore, we have substantially rearranged the main text (from 3,813 words to 2,795 words) and the figures (five Figures and ten Extended Data Figures) by removing repetitive and unessential statements without changing the content of the original manuscript, so the revised manuscript now fits the article format of *Nature*. We believe that the revised manuscript has become more concise and is now acceptable for publication in *Nature*.

The major format changes are as follows.

Main text: The description about the detailed interactions between Dicer-2 and R2D2 has been moved to the legend of the new Extended Data Fig. 2f. The description about a structural comparison between Dicer-2 and hDicer has been moved to the legend of the new Extended Data Fig. 9c.

Fig. 1: We reduced the sizes of the structural figures to fit the format of *Nature*. The labels, “dsRBD1”, “dsRBD2”, “CdsRBD”, “siRNA-1”, and “siRNA-2” have been changed to “RBD1”, “RBD2”, “CRBD”, “dsRNA”, and “siRNA”, respectively.

Fig. 2: According the comment from Referee #2, we repeated the pulldown experiments, analyzed the fractions by western blotting, and updated Fig. 2b. In addition, we reduced the size of the structural figure (Fig. 2a) to fit the format of *Nature*.

Fig. 3 (Fig. 4 in the original manuscript): According the comment from Referee #3, we added a structural comparison of Dicer-2 with the DCL3–dsRNA complex, which was recently reported (Wang *et al.*, *Science*, 2021), and removed the comparison with Drosha. The previous Fig. 3 (the helicase domains of Dicer-2 and RIG-I) has been moved to Extended Data Fig. 6.

Fig. 4 (Fig. 5 in the original manuscript): According the comment from Referee #1, we repeated crosslinking experiments using the K98A mutant, and updated Fig. 4f (the previous Fig. 5g). The previous Fig. 5b (the R2D2–siRNA structure in a ribbon representation) and Fig. 5c (the R2D2–siRNA structure in a surface representation) have been merged into the new Fig. 4b.

Fig. 5 (Fig. 7 in the original manuscript): We reduced the figure size to fit the format of *Nature*.

Extended Data Fig. 2: The previous Extended Data Fig. 3a (a density map for the central linker) has been modified and moved to Extended Data Fig. 2d.

Extended Data Fig. 3: The previous Extended Data Fig. 7 (sequence alignments of the central linkers) has been moved to Extended Data Fig. 3.

Extended Data Fig. 4: The previous Extended Data Fig. 3 (density maps for the bound RNAs) has been modified and moved to Extended Data Fig. 4.

Extended Data Fig. 5: The previous Extended Data Fig. 4 (a structural comparison between the binary and ternary complex) has been moved to Extended Data Fig. 5.

Extended Data Fig. 6: The previous Fig. 3 (the helicase domains of Dicer-2 and RIG-I) has been moved to Extended Data Fig. 6 and merged with the previous Extended Data Fig. 5 (a sequence alignment and the previous Dicer-2 structure).

Extended Data Fig. 7: The previous Extended Data Fig. 6 (the Platform-PAZ domains) has been moved to Extended Data Fig. 7. We added the structure of the Platform-PAZ domain of DCL3.

Extended Data Fig. 8: According the comment from Referee #3, we performed a structural comparison of Dicer-2 with DCL1–dsRNA and DCL3–dsRNA, which were reported recently (Wei *et al.*, *Nat. Plant*, 2021, Wang *et al.*, *Science*, 2021), together with a Dicer-2 model predicted by AlphaFold2. We now present the cryo-EM structures of DCL1 and DCL3, and the predicted Dicer-2–dsRNA models in the new Extended Data Fig. 8.

Extended Data Fig. 9: The previous Fig. 6 (a structural comparison between Dicer-2 and hDicer) has been modified and moved to Extended Data Fig. 9.

Extended Data Fig. 10: According to the comments from Referees #1 and #3, we performed crosslinking experiments using siRNA duplexes with a 0–2-nt 3'-overhang. We added the results of the crosslinking experiment and the structural figures of Dicer-2–R2D2–siRNA in the new Extended Data Fig. 10.

Our point-by-point responses to the reviewers' comments are as follows.

Reviewer #1 (Remarks to the Author):

Yamaguchi and co-workers present the structure of the Dcr2-R2D2 heterodimer with and without bound siRNA duplexes. Many in the RNAi field, including myself, have waited many years to see this structure and the results do not disappoint. The asymmetry rule for siRNA guide strand selection was established nearly two decades ago. While several asymmetry sensors have been proposed, the best characterized is the Drosophila Dcr2-R2D2 heterodimer. This complex is also essential for siRNA production in flies and has been an instrumental part of mechanistic studies of RNAi. Yamaguchi et al., now reveal the structure of the heterodimer bound to an siRNA duplex at atomic resolution. The structure provides the most accurate and complete structure of any eukaryotic Dicer protein, the atomic details of asymmetry sensing by R2D2, and insights into dsRNA recognition and processing by the Dicer helicase. The study was a joy to read and, in my opinion, will make a fantastic contribution.

We thank the reviewer for appreciating the importance of the structures of the Dicer-2–R2D2 heterodimer with and without a bound siRNA duplex. We have addressed the points raised by the reviewer as follows.

I have the following questions, primarily surrounding the details of asymmetry sensing:

1) Line 223: “The unstable end interacts with neither Dicer-2 nor R2D2...”. This is surprising considering data in Fig. 5G and previous results (PMID: 15550672) showing the unstable siRNA end efficiently crosslinks to Dcr-2. If asymmetry sensing is entirely through R2D2, as the structure would suggest, why is crosslinking to Dcr2 unaffected by the W205A mutation? Is it possible that Dcr2 does engage the unstable siRNA end, but this conformation of the complex was not captured in the single particle analysis?

Thank you for your helpful comments. Although we stated that “The unstable end interacts with neither Dicer-2 nor R2D2...” in the original manuscript, the siRNA unstable end is located close to the PAZ domain of Dicer-2. In particular, K924 in the PAZ domain may electrostatically interact with the backbone phosphate group between C18 and U19 in the passenger strand, although we did not originally focus on the possible interaction in the original manuscript due to the relatively poor density for the K924 side chain (Fig. L1). Furthermore, the structural comparison between Dicer-2–R2D2 and Dicer-2–R2D2–siRNA implies that the PAZ domain undergoes a conformational change upon siRNA binding. These structural observations suggest the possibility that the siRNA unstable end can be crosslinked to the PAZ domain of Dicer-2. We have therefore removed the statement “The unstable end interacts with neither Dicer-2 nor R2D2...” and added the possible interaction between siRNA and Dicer-2 in the new Fig. 4a in the revised manuscript.

Fig. L1 | Density map for K924 and the siRNA

2) Fig. 5E indicates stability at the very end of the siRNA duplex is detected by stacking of guide strand nucleobase-20 with W205, and passenger nucleobase-1 with the aliphatic portion of K98. Fig 5G. shows guide crosslinking to R2D2 is lost in the W205A mutant. Have the authors determined how the W205A mutation impacts siRNA affinity?

As shown in the new Fig. 4f, the passenger-labeled siRNA duplex (p*) was similarly crosslinked with Dicer-2 regardless of the W205A mutation in R2D2. These results indicate that the W205A mutation does not substantially affect the overall affinity to siRNA duplexes, as the sugar-phosphate backbone interactions with dsRBD1 and dsRBD2 of R2D2 mainly contribute to the siRNA binding *per se*.

Have the authors examined crosslinking in the K98A mutant?

According to the reviewer's comment, we examined the crosslinking of the K98A mutant, and found that, like the wild-type R2D2, the K98A mutant is crosslinked with the siRNA duplex. These results indicated that the K98A mutation does not substantially affect the siRNA binding, since the siRNA also interacts with other residues of R2D2. We have added these results in the revised manuscript (new Fig. 4f).

3) The structure shown in Fig. 5E indicates RD2D actually only recognizes one nucleotide in the siRNA 3' overhang (as nucleotide 21 is disordered). Does biochemistry support this mechanism? i.e. would a shortened siRNA, with a single nucleotide 3' overhang, crosslink equally well or is a 2 nucleotide overhang needed?

We appreciate the Reviewer's constructive comment. Accordingly, we performed crosslinking experiments using siRNA duplexes with a 0–2-nt 3'-overhang at their stable ends. Intriguingly, all of the siRNA duplexes were similarly crosslinked to R2D2 (new Extended Data Fig. 10c), suggesting that R2D2 does not recognize the overhang length at the siRNA stable end. Supporting this observation, the terminal base pair of a manually modeled blunt-end siRNA duplex (0-nt 3'-overhang) stacks with W205 and K208 of R2D2 (new Extended Data Fig. 10d). We have added these results in the revised manuscript.

4) Line 248: “the W205A mutation and the CTD α 2 deletion abolished the crosslinking of g to R2D2..... indicating that the siRNA stable end does not form a stacking interaction with these R2D2 mutants.” This sentence is confusing....how do the authors know crosslinking correlates with stacking in the CTD α 2 deletion? For the W205A mutant, how could stacking interactions be formed in the first place?*

Thank you for your critical comment. The observations that the W205A mutation and the CTD α 2 deletion abolished the crosslinking of g* to R2D2, but not that of p* to Dicer-2, indicate that the nucleotide g20 was crosslinked to R2D2 via W205, consistent with our structural finding that g20 in the siRNA stacks with W205 of R2D2. Nonetheless, as the reviewer pointed out, the W205A and the CTD α 2 deletion mutants of R2D2 cannot be crosslinked to the siRNA duplex, since they lack an aromatic residue (W205) that can crosslink to the 5-iodouracil in the siRNA duplex. Thus, we have removed the statement “the W205A mutation and the CTD α 2 deletion abolished the crosslinking of g* to R2D2..., indicating that the siRNA stable end does not form a stacking interaction with these R2D2 mutants.” in the revised manuscript.

5) Fig. 3E: why are the two strands designated “guide” and “passenger”? Is this a typo or are the authors suggesting that orientation bound to the helicase can determine which strand will be loaded as guide? I ask because such a notion is difficult to reconcile with the model for siRNA release and rebinding after dicing. Also, considering the limited quality of the density in Extended Data Fig. 3B, I am wondering how the authors can know which end of the siRNA duplex is bound.

Thank you for your helpful comment. As the reviewer pointed out, we cannot distinguish between the guide and passenger strands in the siRNA duplex bound to the helicase domain (siRNA-1). Thus, we have decided not to distinguish between the two strands in siRNA-1 and now refer to “siRNA-1” and “siRNA-2” as “dsRNA” (gray colored, rather than red/blue) and “siRNA” in the revised manuscript, respectively.

6) *The mechanism for Ago-loading in Supplementary Movie 2 is intriguing because it predicts the relative positions of Dicer2 and Ago2. Sasaki and Shimizu identified an Ago-binding motif in the RNaseIIIa domain of human dicer many years ago (PMID: 17482383). Is the equivalent region of DmDcr2 near the surface predicted to interact with Ago2?*

A sequence comparison indicated that the identified Ago-binding motif of hDicer corresponds to the RIIIi domain of Dicer-2. Consistently, the RIIIi domain of Dicer-2 is located near the MID-PIWI domains of Ago2 in our docking model (Extended Data Movie 2). Nonetheless, we did not include the comparison, since it remains unclear whether hDicer–TRBP mediates siRNA loading into Ago2 in the identical manner to that of Dicer-2–R2D2.

7) *Line 147: “...that central linker (residusses 1081–1115) interacts with...”. Residues is misspelt, but more importantly, the residue numbers seem incorrect.*

Thank you for the comment. We have corrected it.

8) *Supplementary Movie 1: the central linker (CL) is mislabeled “CH”.*

Thank you for the comment. We have amended it.

Referee #2 (Remarks to the Author):

This study provides cryo-EM structures of fly Dicer-2-R2D2 heterodimer and Dicer-2-R2D2-siRNA complex at high resolutions. These structures contain previously unidentified regions including a linker between the Platform-PAZ and RIII domains (“central linker”) and a large α -helical domain within the RIIIa domain (“RIIIi”). These structures of these domains are difficult to predict using AlphaFold2 and turn out to form extensive contacts with other domains of Dicer-2 and R2D2. The authors captured two different binding modes of siRNA substrates, presumably indicating the initial recognition state and pre-loading state, respectively. In particular, the pre-loading state structure provides atomic-level insights into how Dicer-2 and R2D2 sense siRNA thermodynamic asymmetry, for the first time. Although a low-resolution structure (7.1 Å) of fly Dicer-2-R2D2 complex with dsRNA bound to the helicase domain (which represents the pre-Dicing state) was previously reported (Sinha et al., 2018), the current study substantially advances our understanding of small RNA biology by solving a high-resolution (3.3 Å) structure in the pre-loading state.

We appreciate the positive comments from the reviewer. We have addressed the points raised by the reviewer as follows.

Major comments:

1. The authors show, in the structure and with photo-crosslinking experiments, that R2D2 with specific amino acids interacts with the central region of the siRNA duplex and forms a stacking interaction with the base of g20. However, the major difference between the stable and unstable ends of siRNA duplex lies in the Watson-Crick base-pairing at the end of the stem (g19) between the two strands. There is no clear structural explanation of how R2D2 senses such base-pairing difference. Therefore, building structural models with the unstable end positioned in the pocket formed by the dsRBD2 and CTD of R2D2 might help get insights to explain the asymmetry mechanism much clearer.

We appreciate your helpful comment. Accordingly, we constructed a structural model, in which the siRNA unstable end is located near R2D2 (Fig. L2). The structural model suggested that, like the siRNA stable end, the siRNA unstable end is also capable of binding R2D2 even with the g1-p1 mismatch. Thus, R2D2 does not directly distinguish the presence or absence of the Watson-Crick base-pairing at the end of the stem. Indeed, previous biochemical analyses have demonstrated that R2D2 senses the thermodynamic asymmetry of fully paired siRNA duplexes that lack any mismatches (Tomari *et al.*, *Science*, 2004). Accordingly, we reasoned that the strand selection mechanism by R2D2 occurs as follows: R2D2 prefers to bind the double-helical conformation at the end of an siRNA duplex in a sequence-independent manner. Consequently, the more thermodynamically stable end (with greater double-helical character) of the siRNA duplex is preferentially anchored by R2D2 in equilibrium, leading to the asymmetric recognition of the siRNA duplex by Dicer-2–R2D2.

Fig. L2 | Structural model, in which the siRNA unstable end is located near R2D2

2. They argued that W205 and CTD α 2 of R2D2 are critical for the siRNA thermodynamic asymmetry based on the photocrosslinking assay (Figure 5G). However, in contrast to the deletion of CTD α 2, the W205A mutation didn't show a significant increase of g^ binding. Is it possible that the W-to-A*

mutation just abolishes the crosslinking not the asymmetric binding? If it is difficult to exclude this possibility, please focus on the importance of CTD α 2.

Thank you for your critical comment. We fully agree that the W205A mutant abolished the siRNA crosslinking, but not the asymmetric binding. Thus, we have decided to focus on the importance of the CTD α 2 for the siRNA thermodynamic asymmetry sensing in the revised manuscript.

3. The authors suggested that the central linker (1081-1115 aa) has amino acid residues specifically interacting with R2D2 (Figure 2A). These amino acids include Y1089, Y1103, V1105, and V1109. However, the pulldown experiments show that FLAG-tagged R2D2 has no problem pulling down Dicer-2 mutant (Δ 1082-1115) that lacks all four amino acid residues above. Although a smaller amount of Dicer-2 mutant was pulled down compared to when wild-type Dicer-2 was expressed, but this is likely because of the low expression levels of Dicer-2 mutant (as observed when pulled down with a His tag). It would be great if the authors can elaborate more on this, whether the interaction would be, for example, meaningful in the context of processing substrates.

Thank you for your critical comment. We repeated the pulldown experiments and carefully analyzed the fractions by western blotting with anti-FLAG and anti-Dicer-2 antibodies (rather than CBB staining as in the original manuscript). We found that the Dicer-2 Δ 1082-1115 mutant binds to R2D2 less efficiently as compared to the wild-type Dicer-2, while their expression levels are similar. These results confirmed the contribution of the central linker to R2D2 binding. We show the result as the new Fig. 2b in the revised manuscript.

4. As mentioned in the paper, “Dicer-2, but not Dicer-2-R2D2, can process pre-miRNAs, indicating that R2D2 acts in determining the specificity of Dicer-2 for the dsRNA substrates” (there is no citation, by the way). According to the model of siRNA production by Dicer-2-R2D2 proposed in Figure 7, Dicer-2 recognizes a long dsRNA substrate via the helicase domain. In the structure, however, R2D2 is isolated from where initial substrate recognition is made and it is difficult to imagine how R2D2 can block pre-miRNA processing, especially when the helicase domain recognizes not only blunt ends but also 2-nt 3'-overhang (as observed in the structure). Although the authors illustrated how R2D2 might sterically clash with a pre-miRNA that enters from the center, it would be great to know how the authors perceive this scenario.

Thank you for your helpful comment. Cenik *et al.* reported that pre-miRNAs are inaccurately processed by Dicer-2, but not Dicer-2-R2D2, indicating that R2D2 inhibits promiscuous pre-miRNA processing by Dicer-2 (Cenik *et al.*, *Mol. Cell*, 2011). We have cited this reference in the revised manuscript. Previous studies indicated that hDicer cleaves pre-miRNA substrates in an ATP-

independent manner, whereas Dicer-2 cleaves long dsRNA substrates in an ATP-dependent manner (Cenik *et al.*, *Mol. Cell*, 2011, Ma *et al.*, *J. Mol. Biol.*, 2008). Whereas hDicer recognizes a pre-miRNA substrate using both the PAZ and helicase domains (Liu *et al.*, *Cell*, 2018), Dicer-2 initially recognizes a long dsRNA substrate using only the helicase domain (Sinha *et al.*, *Science*, 2018) (Fig. L3). Based on the hypothesis that, like hDicer, Dicer-2 recognizes a pre-miRNA using both the PAZ and helicase domains, we performed a structural comparison between Dicer-2–R2D2 and hDicer–TRBP–pre-miRNA. The modeled pre-miRNA substrate sterically clashes with R2D2, thereby explaining why R2D2 inhibits promiscuous pre-miRNA processing by Dicer-2. We have added these explanations in the revised manuscript (the legend of Extended Data Fig. 9c).

Fig. L3 | Substrate RNA recognition by hDicer and Dicer-2

Minor comments:

1. Please mark the CTD $\alpha 2$ in Figure 5B in addition to Figure 5E, which would help the understanding.

According to the reviewer’s comment, we have modified Fig. 5b (new Fig. 4b) in the revised manuscript.

2. Typos

- Line 103, reference 37 → reference 30 (and its reference style also differs from the others)
- Lines 111 and 115, Figures 2X → Extended Data Figures 2X (?)
- Line 146: residusses

Thank you for the comment. We have corrected them.

Referee #3 (Remarks to the Author):

In this manuscript, the authors have solved the cryoEM structures of the Drosophila Dicer2-R2D2 complex and the Dicer2-R2D2-ds-siRNA complex. The resolution is 3.3 Å and allows for building de novo atomic models. The structures are in a post-cleavage conformation where the siRNA is stably bound by R2D2. This is expected since a short RNA has been included and cleavage should not occur. With their structures, the authors confirm and validate several mechanistic aspects that have been suggested before based on biochemical experiments. In addition, there are also several novel features. Comparing the two complexes, the authors revealed that the dsRBd1 of R2D2 becomes ordered when it binds to the central region of the siRNA duplex. The interaction surface of Dicer-2 and R2D2 is also further refined on an atomic level. Comparison of the helicase domains of Dicer-2 and RIG-I also allows for conclusions on dsRNA recognition of this domain although the bound RNA is not well resolved and thus only modeled. The authors also compare the active site and the 5' and 3' binding pockets with a human cryoEM structure, that is in a different conformation. The authors present several mechanistic assumptions based on the two different conformations. Finally, the authors analyzed the strand selection by R2D2 and further confirm that R2D2 anchors the siRNA in an orientation in which the less stable end is exposed to the solvent and the more stable end is anchored on R2D2. This is a very clear, well written manuscript presenting an important structure of the Dicer-2-R2D2-siRNA complex that deserves publication in such a top journal. The structure allows for a better understanding of the mechanisms underlying strand selection and subsequent siRNA loading into Ago. However, many of the observed features confirm known models and several aspects are solely based on modeling and assumptions. Listed below are a number of points/issues that should be clarified or discussed.

We appreciate the positive comments by the reviewer. We have addressed the points raised by the reviewer as follows.

1. On page 5, lines 128-130: the authors report that the density of the siRNA bound to the helicase domain was relatively unclear suggesting flexibility. Thus, they modeled the unstable end of siRNA-1 according to the shapes of the densities. This needs to be explained better. If the densities are so clear that the unstable end can be modeled, then it is not clear why the siRNA cannot be better defined. Otherwise the modeling is rather random and might not be a very strong basis for the mechanistic conclusions that are drawn later when comparing it with the RIG-I structure. Particularly, the conclusions on blunt end and overhang binding seems to be rather speculative. Extended data Figure 5D is not mentioned in the text.

Thank you for your critical comment. Although the density of the siRNA duplex bound to the helicase domain was relatively unclear, it was fitted better to the unstable end than the stable end of the RNA duplex (Extended Data Fig. 4b). Thus, we modeled nucleotides at the unstable end (g1–5 and p15–21) into the density. Accordingly, we now refer to “siRNA-1” as “dsRNA”, and do not discriminate between the two strands of the dsRNA bound to the helicase domain in the revised manuscript. We added these explanations and cited Extended Data Fig. 5d (new Extended Data Fig. 6h) in the revised manuscript.

2. Page 5, bottom: R2D2 dsRBD1 is disordered in the siRNA unbound state. What is the evidence that it is disordered? If the correctly folded domain would be highly mobile and flexible in the unbound state, it would also not be resolved in the structure. Or are there other data suggesting that it is disordered?

As shown in Extended Data Fig. 1g, h, the density of dsRBD1 was observed in the Dicer-2–R2D2–siRNA structure, but not the Dicer-2–R2D2 structure, suggesting that dsRBD1 is highly mobile and flexible in the siRNA-unbound state. Consistent with this notion, dsRBD1 and dsRBD2 of R2D2 are connected by a flexible linker. We have modified the main text and cited Extended Data Fig. 1g and 1h (density maps for Dicer-2–R2D2 and Dicer-2–R2D2–siRNA) in the revised manuscript.

3. On Page 7: the authors model long dsRNA into their structure and find that it clashes at various positions. This leads to the conclusion that there must be mechanistically important rearrangements during siRNA production. Although this is likely true, the conclusion from the data is somewhat far-fetched based on a comparison of hDicer and Drosophila Dicer-2. I think the final statement that “...our structure provides mechanistic insights into Dicer-2 catalyzed high-fidelity siRNA production” is not justified and should be rephrased and toned down.

We agree with the reviewer that it is somewhat far-fetched to discuss the structural rearrangement of Dicer-2 during siRNA production, solely based on the structural comparison between hDicer and Dicer-2. However, two cryo-EM structures of the plant Dicer proteins (DCL1 and DCL3) bound to their dsRNA substrates were recently reported, elucidating the dsRNA recognition mechanisms by the Dicer enzymes (Wei *et al.*, *Nat. Plant*, 2021, Wang *et al.*, *Science*, 2021). By comparing the structures of Dicer-2, DCL1, DCL3, and an AlphaFold2-predicted Dicer-2 model, we obtained mechanistic insights into dsRNA recognition by Dicer-2, including the rearrangement in the helicase domain (new Fig. 3, Extended Data Figs. 7 and 8). We have added these results in the revised manuscript and toned down the statement about the siRNA production mechanism, according to the reviewer’s suggestion.

4. Page 8, last sentence of second paragraph: “A modeled blunt siRNA sterically clashes...”. This is unclear. First, ‘data not shown’ should be added. Second, isn’t a blunt end siRNA duplex also shown in 5E? And why would a blunt end clash while overhangs not? This is confusing and should be described clearer.

We appreciate the critical comments. We have now directly addressed this point by performing crosslinking experiments using siRNA duplexes with a 0–2-nt 3'-overhang at their stable ends. Intriguingly, all of the siRNA duplexes were similarly crosslinked to R2D2 (new Extended Data Fig. 10c), indicating that R2D2 does not distinguish the overhang length at the siRNA stable end and can bind to a blunt-end siRNA duplex. A modeled blunt-end siRNA duplex supports this idea, although it appears to slightly clash with the side chain of K208 (new Extended Data Fig. 10d). These results suggest that the side chain of K208 undergoes a conformational change upon the binding of a blunt-end siRNA duplex. We have added the results of the crosslinking experiments and the structural model in the revised manuscript.

5. Figure 5F: the authors state several times that the less stable end is exposed to the solvent. On the other hand, both ends can be cross linked to the complex. This should be consistent and described clearer.

Thank you for your critical comment. Although we stated that “The unstable end interacts with neither Dicer-2 nor R2D2...” in the original manuscript, the siRNA unstable end is located close to the PAZ domain of Dicer-2 (related to the response to Reviewer #1). In particular, K924 in the PAZ domain may electrostatically interact with the backbone phosphate group between C18 and U19 in the passenger strand, although we did not originally mention this possible interaction due to the relatively poor density for the K924 side chain. Furthermore, a structural comparison between Dicer-2–R2D2 and Dicer-2–R2D2–siRNA implied that the PAZ domain undergoes a conformational change upon siRNA binding. These structural observations suggest the possibility that the siRNA unstable end can be crosslinked to the PAZ domain of Dicer-2. Accordingly, we have removed the statement “The unstable end interacts with neither Dicer-2 nor R2D2...”, and added the explanation in the revised manuscript.

6. One of the most intriguing question in this context is how the more stable and the less stable ends are sensed. I understand that there is some kind of dissociation and association and during this step the correct orientation is achieved. But how is it sensed? This is not really addressed in this study. The conclusions made here seem to be mainly confirmative of already proposed models. Are there specific residues that would prefer G-C pairs over A-U? Or are there sterical features that would

require a stable structure and less 'breathing' of the end? Maybe the analyses could be extended into this direction.

We fully agree with the reviewer that an siRNA duplex binds to R2D2 in the correct orientation in equilibrium (during dissociation and association). Based on our structural and functional data, we reason that the strand selection mechanism by R2D2 is as follows: R2D2 prefers to bind the double-helical conformation at the end of an siRNA duplex in a sequence-independent manner.

Consequently, the more thermodynamically stable end (with greater double-helical character) of the siRNA duplex is preferentially anchored by R2D2 in equilibrium, leading to asymmetric recognition of the siRNA duplex by Dicer-2–R2D2.

7. Reference to extended data Figures 1 and 2 are not accurate in the first two paragraphs of the results section. This should be revisited and corrected.

Thank you for the comment. We have corrected it.

Reviewer Reports on the First Revision:

Referees' comments:

Referee #1 (Remarks to the Author):

The authors have addressed my questions and concerns. In my opinion, the revised manuscript has benefitted from the review process. An explanation for how R2D2 discourages processing of pre-miRNAs was a particularly nice addition. I have two more comments, which are entirely cosmetic and only suggestions for what is already a clear and thought-provoking manuscript:

Line 147: "RIG-I contains the CTD domain, which is connected to the Hel2 domain with the V-shaped pincer helix (Extended Data Fig. 6c)." The term 'CTD' here is potentially confusing because 'CTD' is also used to describe the C-terminal domain of R2D2. Consider instead using the term 'C-terminal regulatory domain' for referring to the RIG-I domain.

As a reader, I found it a little confusing to display the two duplexes in Fig. 4e in opposite orientations. I recognize that this arrangement helps the reader understand how the duplexes were prepared for the experiment (the authors cleverly used a single modified oligonucleotide as both guide and passenger by changing the sequence of the unmodified oligo). However, for me, the experimental setup would have been easier to absorb had both duplexes been presented in the same orientation with respect to functionality (i.e. guide strand always above the passenger).

Referee #2 (Remarks to the Author):

The authors have clarified all of these issues I had raised and explained their analysis in more detail where necessary. They responded adequately to my review and thus I would be satisfied with the new version of the manuscript.

Referee #3 (Remarks to the Author):

In the revised version of their manuscript, the authors have addressed all points that I had raised on the previous version. The authors have adequately responded to my comments and therefore I am satisfied with the response.

Author Rebuttals to First Revision:

Our point-by-point responses to the reviewers' comments are as follows.

Referees' comments:

Referee #1 (Remarks to the Author):

The authors have addressed my questions and concerns. In my opinion, the revised manuscript has benefitted from the review process. An explanation for how R2D2 discourages processing of pre-miRNAs was a particularly nice addition. I have two more comments, which are entirely cosmetic and only suggestions for what is already a clear and thought-provoking manuscript:

Thank you for your favorable comment.

Line 147: "RIG-I contains the CTD domain, which is connected to the Hel2 domain with the V-shaped pincer helix (Extended Data Fig. 6c)." The term 'CTD' here is potentially confusing because 'CTD' is also used to describe the C-terminal domain of R2D2. Consider instead using the term 'C-terminal regulatory domain' for referring to the RIG-I domain.

Thank you for the helpful suggestion. According to the comments, we used the term "carboxy-terminal regulatory domain (CRD)" for referring to the RIG-I CTD.

As a reader, I found it a little confusing to display the two duplexes in Fig. 4e in opposite orientations. I recognize that this arrangement helps the reader understand how the duplexes were prepared for the experiment (the authors cleverly used a single modified oligonucleotide as both guide and passenger by changing the sequence of the unmodified oligo). However, for me, the experimental setup would have been easier to absorb had both duplexes been presented in the same orientation with respect to functionality (i.e. guide strand always above the passenger).

Thank you for the suggestions. According to the comments, we displayed two duplexes in Fig. 4e in the same direction.

Referee #2 (Remarks to the Author):

The authors have clarified all of these issues I had raised and explained their analysis in more detail where necessary. They responded adequately to my review and thus I would be satisfied with the new version of the manuscript.

Thank you for your favorable comment.

Referee #3 (Remarks to the Author):

In the revised version of their manuscript, the authors have addressed all points that I had raised on the previous version. The authors have adequately responded to my comments and therefore I am satisfied with the response.

Thank you for your favorable comment.

Thank you again for your thoughtful consideration. We hope that the revised manuscript is now acceptable for publication in *Nature*.